Brief Communication

# A combinatorial genetic strategy for exploring complex genotype–phenotype associations in cancer

Shan Li[1,12], Alicia Wong[1,12], Huiyun Sun[1,2,12], Vipul Bhatia[1], Gerardo Javier[1], Sujata Jana [1], Qian Wu[3,4], Robert B. Montgomery[5], Jonathan L. Wright[6], Hung-Ming Lam[6], Andrew C. Hsieh [1,3,5], Bishoy M. Faltas [7,8,9,10], Michael C. Haffner[1,3,11] & John K. Lee [1,3,5,11] ✉

Available genetically defined cancer models are limited in genotypic and phenotypic complexity and underrepresent the heterogeneity of human cancer. Here, we describe a combinatorial genetic strategy applied to an organoid transformation assay to rapidly generate diverse, clinically relevant bladder and prostate cancer models. Importantly, the clonal architecture of the resultant tumors can be resolved using single-cell or spatially resolved next-generation sequencing to uncover polygenic drivers of cancer phenotypes.

Most cancers are not driven by a single oncogenic driver but are instead the sum of multiple genetic perturbations that occur during tumor evolution[1]. However, the functional impact of most genomic abnormalities found in cancers remains largely unknown. Wrangling the catalog of recurrent genetic alterations in cancer and deriving meaningful insights into the functional and contextual contributions of these events is a major challenge in the field of cancer genomics. In vitro assays recapitulate only specific aspects of cancer behaviors such as cell proliferation, anchorage-independent colony formation or invasive migration. In vivo strategies such as the transplantation of cancer cell lines or chemical carcinogenesis are not genetically defined. Genetically engineered mouse models are a gold standard to define genetic drivers in cancer, but they are costly, slow and do not allow the facile manipulation of more than a few genes. Dissociated-cell tissue recombination and transplantation assays have also been applied to study the malignant transformation of primary epithelial cells but have been reliant on the introduction of discrete sets of candidate genes and limited by inefficient transgenesis. Collectively, existing cancer models generated through these methods grossly underrepresent the diversity of human cancer. Furthermore, the use of these technologies to systematically investigate the permutations of genetic events associated with a single cancer would be incredibly challenging, if not impossible. To address these limitations of scale, throughput and economy, we developed a methodology incorporating barcoded lentiviral libraries encoding cancer-associated genetic events introduced efficiently and at random into primary epithelial cells, which are engrafted in mice for tumorigenic selection, at a high multiplicity of infection (MOI). This system enables the generation of genotypically and phenotypically diverse tumors and the massively parallel single-cell lentiviral barcode sequencing of tumors to identify cooperative oncogenic drivers of malignant transformation and specific cancer phenotypes.

Organotypic or organoid cultures permit the expansion of primary epithelial cells while maintaining their complex organization and tissue function[2]. A major barrier to higher-order genetic studies in this context has been inefficient transgenesis using available lentiviral transduction protocols. We proposed that enforced cell–virus contact

[1]Human Biology Division, Fred Hutchinson Cancer Center, Seattle, WA, USA. [2]Molecular Engineering and Sciences Institute, University of Washington, Seattle, WA, USA. [3]Clinical Research Division, Fred Hutchinson Cancer Center, Seattle, WA, USA. [4]Public Health Sciences Division, Fred Hutchinson Cancer Center, Seattle, WA, USA. [5]Division of Medical Oncology, University of Washington School of Medicine, Seattle, WA, USA. [6]Department of Urology, University of Washington School of Medicine, Seattle, WA, USA. [7]Sandra and Edward Meyer Cancer Center, Weill Cornell Medicine, New York, NY, USA. [8]Caryl and Israel Englander Institute for Precision Medicine, Weill Cornell Medicine, New York, NY, USA. [9]Department of Medicine, Weill Cornell Medicine, New York, NY, USA. [10]Department of Cell and Developmental Biology, Weill Cornell Medicine, New York, NY, USA. [11]Department of Pathology and Laboratory Medicine, University of Washington School of Medicine, Seattle, WA, USA. [12]These authors contributed equally: Shan Li, Alicia Wong, Huiyun Sun. ✉e-mail: jklee5@fredhutch.org

in a constrained volume of gel matrix could increase lentiviral transduction efficiency. Primary mouse bladder urothelial (mBU) and prostate epithelial (mPE) cells were isolated by fluorescence-activated cell sorting (FACS) on the basis of a lineage-negative (Lin⁻) (CD45⁻CD31⁻Ter119⁻), EpCAM⁺CD49fhigh immunophenotype (Extended Data Fig. 1a), as these populations self-renew at high frequencies[3] and readily establish organoids in culture (Extended Data Fig. 1b). Cells were mixed into cold Matrigel containing concentrated lentivirus expressing GFP before the seeding and polymerization of organoid droplets[4]. Near complete transduction of mBU and mPE cells was achieved, delivering up to 10–20 copies per cell (Fig. 1a,b). We next developed a barcoding system to characterize the distribution of unique proviral copies per cell. Lentiviral constructs were barcoded with matching ten-nucleotide sequences distal to the 5′ long terminal repeat (LTR) and proximal to the 3′ LTR and produced as a pool. A custom single-cell amplicon panel was designed on the Mission Bio Tapestri platform to enable the sensitive enumeration of multiple uniquely barcoded lentiviruses per cell (Extended Data Fig. 1c). This approach was validated using a defined population of 3T3 cells engineered with lentiviruses to harbor up to four unique lentiviral barcodes per cell (Extended Data Fig. 1d). mPE were transduced with a diverse barcoded lentiviral pool at varying MOIs, and single-cell amplicon sequencing showed relatively normal distributions of proviral copies per cell (Fig. 1c and Extended Data Fig. 1e).

To determine the utility of this approach in understanding the initiation and progression of bladder and prostate cancer, we selected commonly mutated genes from cancer genome sequencing studies[5–7] (Extended Data Fig. 2a) and cloned these as open reading frames (ORFs) or short hairpin RNAs (shRNAs) into barcoded lentiviral constructs to mimic gain-of-function or loss-of-function events (Extended Data Fig. 2b and Supplementary Table 1). At least three shRNAs from The RNAi Consortium (TRC) targeting each gene were tested for knockdown in 3T3 cells by quantitative PCR. The shRNA demonstrating the most potent knockdown of target gene expression was incorporated into the lentiviral libraries (Extended Data Fig. 2c). A bladder urothelial lentiviral pool (BU-LVP) of 33 genes and a prostate epithelial lentiviral pool (PE-LVP) of 24 genes were produced in arrayed format to avoid lentiviral barcode recombination and concentrated by ultracentrifugation (Extended Data Fig. 3a). Infectivity (representation) of each lentivirus was evaluated by transducing either mBU or mPE cells with the respective lentiviral pool and performing bulk amplicon sequencing of lentiviral barcodes (Extended Data Fig. 3b). Initial lentivirus pools demonstrated over tenfold overrepresentation of shRNA vectors relative to ORF vectors (Extended Data Fig. 3c), presumably owing to more efficient viral packaging because of the reduced length between LTRs of the transfer plasmid[8]. These data were applied to adjust the cell surface area of producer cells for subsequent arrayed lentiviral library production, leading to near normalization of the representation of shRNA and ORF vectors (Extended Data Fig. 3d).

We adopted an approach in which primary mBU and mPE cells infected with BU-LVP or PE-LVP at high MOI in organoids were recombined with inductive mouse embryonic day 16 (E16) bladder mesenchyme (EBLM)[9] or urogenital sinus mesenchyme (UGSM)[10] and subsequently grafted subcutaneously in NOD scid gamma (NSG) mice to enable biological selection for tumorigenic clones (Fig. 1d). No tumors were appreciable from control grafts of untransduced mBU or mPE cells recombined with EBLM or UGSM. The efficiency of tumor formation (tumors formed per graft inoculated) was 80% (16 of 20) for mBU cells infected with BU-LVP and 38% (18 of 47) for mPE cells infected with PE-LVP (Supplementary Table 2). Tumor latency was measured as time from inoculation to achieving a maximal tumor diameter of 1 cm and ranged from 2.3 to 7.4 months (mean 4.2 months) for bladder tumors and 3.2 to 16 months (mean 8.9 months) for prostate tumors (Supplementary Table 2).

A representative tumor derived from primary mBU cells transduced with BU-LVP exhibited three morphologically distinct regions consistent with papillary urothelial carcinoma with an inverted growth pattern, urothelial carcinoma with squamous differentiation and sarcomatoid urothelial carcinoma, all three of which were also supported by GATA3, TP63 and pan-cytokeratin (panCK) immunostaining (Fig. 1e). Single-cell DNA amplicon sequencing was performed to enumerate the lentiviral barcodes for the determination of clonal architecture and deconvolution of lentivirus-delivered genetic events putatively involved in tumorigenesis. Three major clones harboring distinguishable sets of lentiviral barcodes were identified (Fig. 1f), but spatial resolution was lost owing to single-cell dissociation. To associate histology with clonality, we performed laser capture microdissection (LCM) of the three regions on stained tissue sections and performed bulk DNA amplicon sequencing (Fig. 1g). The papillary urothelial carcinoma was uniquely associated with *Fgfr3* S243C, sh*Atm* and *Zfp703* mutations, in addition to the common *Ywhaz*, *Pik3ca* E545K, *Pparg* and *Pvrl4* mutations observed in all three dominant clones. Cancer genomics studies have shown that activating mutations in *FGFR3* are highly enriched in papillary urothelial carcinomas[5,11]. We further validated these findings in the mouse urothelial transformation assay in independent experiments using a defined lentiviral pool of *Fgfr3* S243C, *Ywhaz*, *Pik3ca* E545K, *Pparg* and *Pvrl4* (Extended Data Fig. 4a), which produced tumors with papillary urothelial carcinoma with an inverted growth pattern by histopathology and based on the endophytic proliferative pattern (Extended Data Fig. 4b). The co-occurrence of these genetic alterations was also evident in the human muscle-invasive bladder cancers from The Cancer Genome Atlas bladder cancer (TCGA-BLCA) cohort[5] (Extended Data Fig. 5).

Several tumors called the Fred Hutch Bladder Tumor (FHBT) series have been generated using this methodology, including those with pure urothelial carcinoma and others with mixtures of histologic subtypes (Fig. 2a and Extended Data Fig. 6). The urothelial origin of these tumors was supported by GFP staining (Fig. 2b–d), which was positive even

**Fig. 1 | Efficient lentiviral transduction of primary epithelial cells at high multiplicity of infection and transformation of urothelial cells to tumors with mixed cancer histologies. a**, Top, schematic of a lentiviral (LV) construct with matching barcodes (BCs) at the 5′ and 3′ ends. Bottom, overview of experiments with LV infection of primary mouse cells in organoid culture and quantification of transduction. Created with BioRender.com. CMV, cytomegalovirus; UBC, ubiquitin C; WVH8, Woodchuck hepatitis virus 8 post-transcriptional regulatory element. **b**, Left, brightfield and GFP images of mouse bladder or prostate organoids 72 h after mock or GFP LV transduction. Scale bar, 400 μm. Right, tables summarizing quantification of LV transduction by flow cytometry and LV copies of GFP (± s.d.) by qPCR. **c**, Left, plot of the distribution of LV copies per mPE cell at different MOIs 72 h after transduction. Right, table summarizing viral copy number (VCN) population frequencies at varying MOIs. The experiment was independently repeated three times with similar results.

**d**, Scheme of the mBU organoid transformation assay to uncover functional genotype–phenotype associations in bladder cancer. Created with BioRender.com. **e**, Left, gross image of a tumor arising from mBU transformed with a BU-LVP. Middle, low-magnification image of the hematoxylin and eosin (H&E)-stained tumor section. Right, high-magnification images of H&E-stained and immunohistochemically stained sections of regions with distinct histologies. Scale bars, 50 μm. Each FHBT model is a unique tumor that is the result of an independent experiment. **f**, Clonal architecture of the three dominant clones in the tumor as determined by Mission Bio Tapestri single-cell analysis of LV barcodes. **g**, Left, tumor tissue section after LCM of the histologically distinct regions. Right, table showing the associations between regional tumor histologies and clones in **f** based on LCM and bulk DNA amplicon sequencing of LV barcodes.

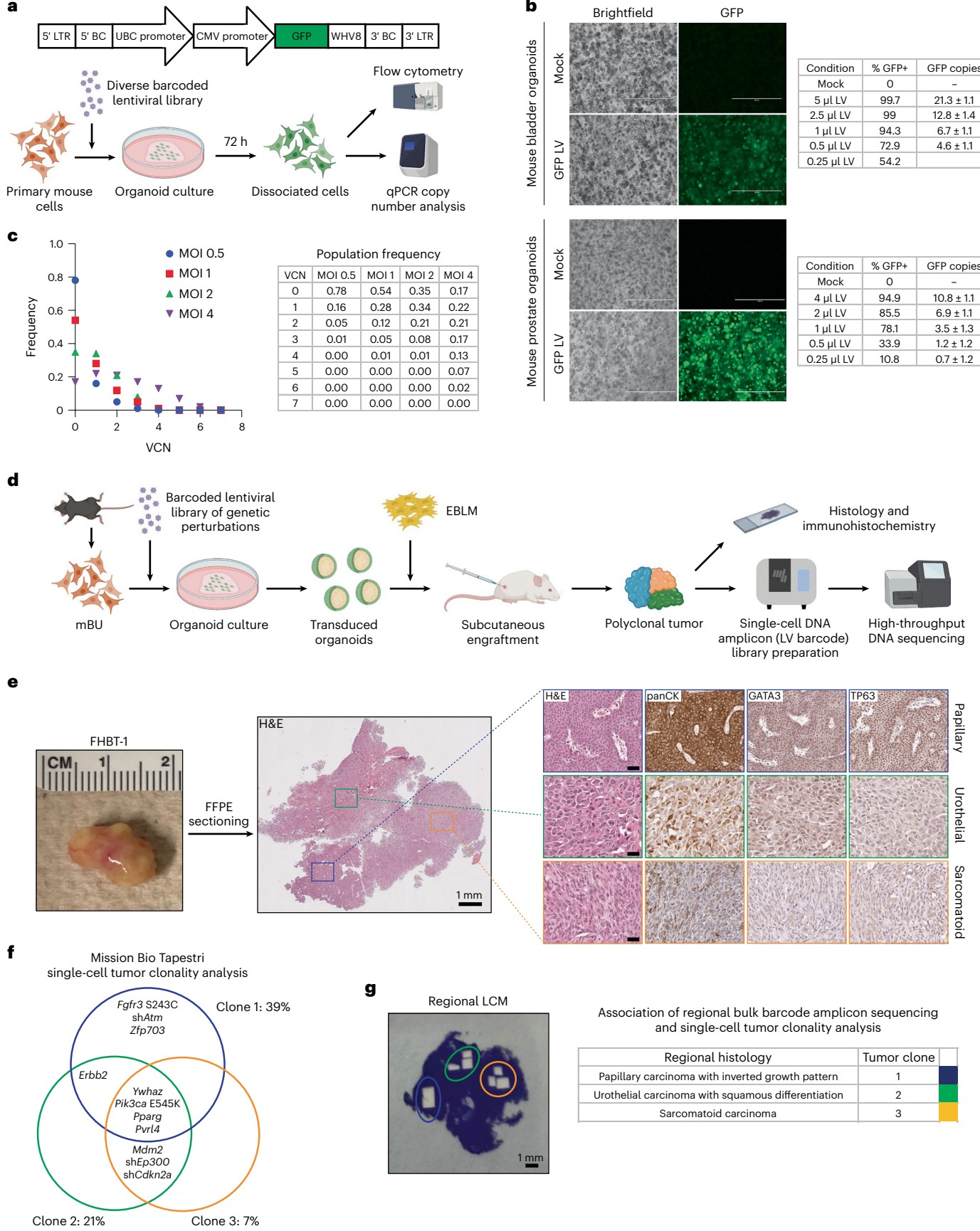

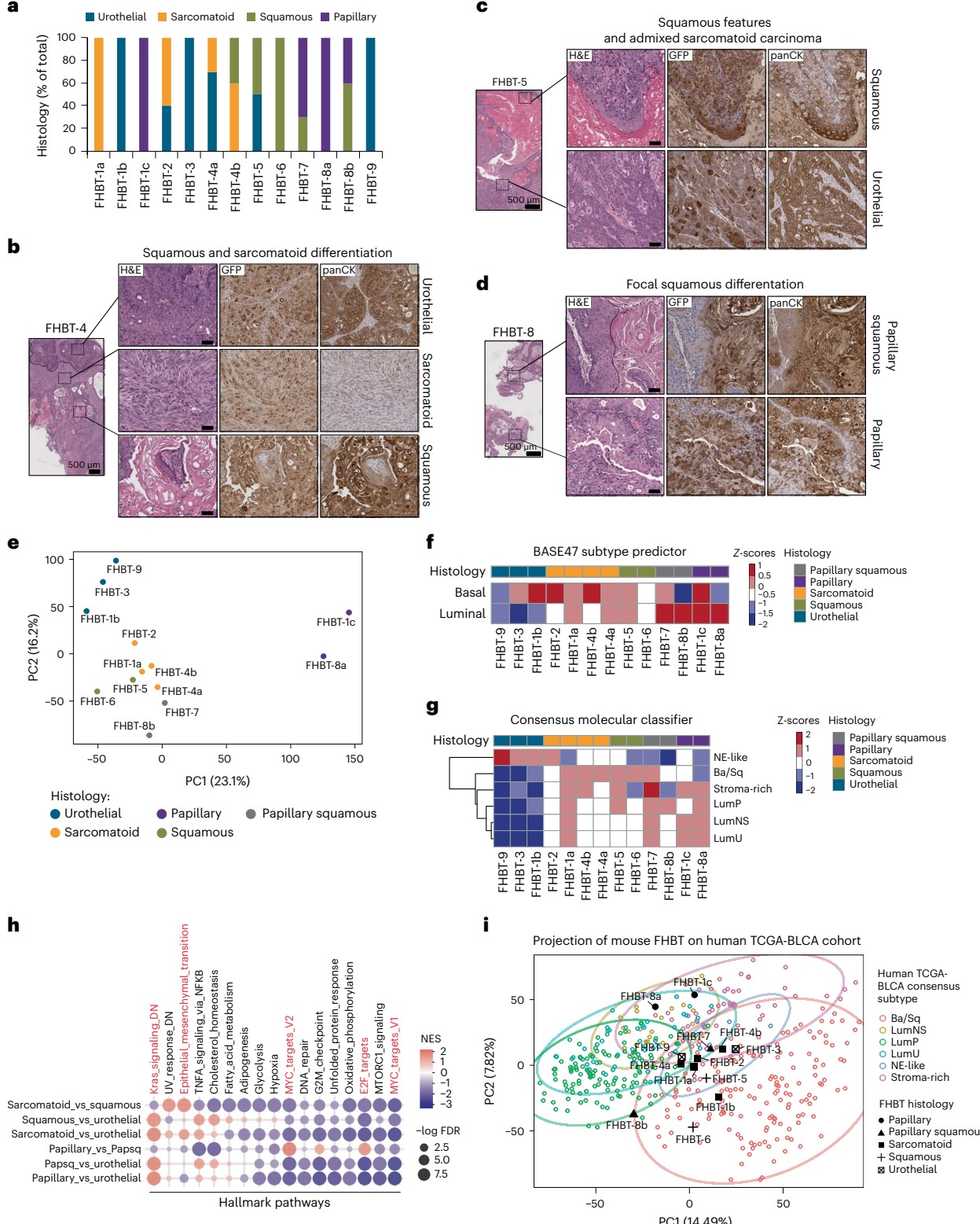

**Fig. 2 | Rapid generation of a series of clinically relevant and phenotypically diverse bladder cancer models. a**, Bar graph showing the representation of cancer histologies present across a series of FHBTs generated using mBU transformed with BU-LVP. **b**–**d**, Low-magnification and high-magnification images of H&E-stained sections and high-magnification images of immunohistochemically stained sections for GFP and panCK expression depicting high-grade urothelial carcinomas with mixed histologies present within the same tumor. Scale bars, 50 μm. Each FHBT model is a unique tumor that is the result of an independent experiment. **e**, PCA plot showing FHBT series color-coded on the basis of histology. Heatmaps showing the histologies of the FHBT series relative to basal and luminal signature scores for the BASE47 subtype predictor (**f**) and signature scores for the Consensus Molecular Classifier (**g**) assigned to neuroendocrine-like (NE-like), basal and/or squamous (Ba/Sq), stroma-rich, luminal papillary (LumP), luminal non-specified (LumNS) and luminal unstable (LumU) subtypes. **h**, Pre-ranked GSEA dotplot of hallmark pathways based on differentially expressed genes (false discovery rate < 0.001) in pairwise histology comparisons. **i**, PCA projection plot of FHBT samples over the TCGA-BLCA samples color-coded by Consensus Molecular Classification (Ba/Sq, LumNS, LumP, LumU, NE-like or stroma-rich) with 90% confidence ellipses shown.

in regions of sarcomatoid carcinoma with low or absent panCK staining (Fig. 2b). We conducted molecular profiling of these tumors and their regional tumor histologies by LCM and RNA-seq analysis. Principal component analysis (PCA) of the gene expression data showed that squamous and sarcomatoid subtypes clustered together and were separate from urothelial and papillary urothelial carcinomas (Fig. 2e). The BASE47 subtype predictor[12], a gene expression classifier used to distinguish luminal and basal subtypes of urothelial carcinoma, generally classified the tumors with papillary and papillary squamous subtypes as luminal and the squamous and sarcomatoid histologies as basal, consistent with an established relationship between sarcomatoid differentiation and the basal subtype[13] (Fig. 2f). The Consensus Molecular Classifier[14] revealed that the non-papillary urothelial histologies showed neuroendocrine-like gene expression with low or absent luminal and basal gene signatures (Fig. 2g and Extended Data Fig. 7a). Gene set enrichment analysis (GSEA) was used to compare these tumor histologies in a pairwise manner and revealed the enrichment of genes associated with epithelial-to-mesenchymal transition in sarcomatoid carcinoma, as expected from prior molecular analyses of human tumors[13] (Fig. 2h). We further confirmed the relevance of our FHBT models by projecting their RNA expression patterns onto principle component analysis plots of tumors from the TCGA-BLCA cohort (Fig. 2i) and established N-butyl-N-(4-hydroxybutyl)-nitrosamine (BBN)-induced mouse bladder cancer models[15] (Extended Data Fig. 7b) to show that they occupy overlapping space on the basis of histologic classification, indicating that the transcriptional features with the greatest variance between tumor subtypes are also conserved with FHBT models.

mPE cells transduced with PE-LVP and engrafted in mice (Extended Data Fig. 8a) also gave rise to mixed cancer morphologies. One tumor showed high-grade prostate adenocarcinoma with focal pleomorphic giant cells (Extended Data Fig. 8b), a rare histologic subtype associated with poor prognosis[16] that may contribute to therapeutic resistance and lethality[17]. Immunostaining revealed HOXB13 and AR expression in both histologies with pronounced nuclear TP53 expression in the pleomorphic giant cells (Extended Data Fig. 8b). We isolated large (pleomorphic giant cell carcinoma) and small (adenocarcinoma) cells from dissociated tumors using a flow cytometry-based strategy, propagated these cells briefly (one passage) in organoid cultures, then dissociated the cells and stained with the nuclear dye Hoechst 33342 to further isolate cells on the basis of DNA content for downstream single-cell lentiviral barcode enumeration (Extended Data Fig. 8c). This single-cell clonality analysis revealed striking enrichment of sh*Kmt2c* in the putative pleomorphic giant cell clones (Extended Data Fig. 8d). Recent studies have established that KMT2C mediates the DNA damage response in cancer[18,19], and DNA damage repair alterations are common in human prostate adenocarcinoma with pleomorphic giant cell features[20].

In summary, we describe a set of technologies that form a functional in vivo cancer genomics assay with efficient delivery of compound genetic perturbations from barcoded lentiviral libraries and single-cell sequencing to rapidly investigate genotype–phenotype relationships in cancer initiation and progression using primary epithelial cells. We leveraged this strategy to develop a series of mouse bladder cancers that recapitulate the phenotypic diversity of human bladder cancer and a mouse prostate cancer with pleomorphic giant cell carcinoma, representing cancer subtypes that have not previously been modeled in a genetically defined fashion. Importantly, single-cell lentiviral barcode deconvolution associated mutant active *Fgfr3* with the luminal papillary differentiation of urothelial carcinoma and the loss of *Kmt2c* with pleomorphic giant cell carcinoma in prostate cancer. These initial studies provide proof of principle that this approach can be deployed to investigate higher-order genetic interactions to explore complex genotype-to-phenotype relationships in cancer.

## Online content

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

## Methods

### Lentiviral constructs and lentiviral library production

Double-barcoded lentiviral vectors were generated from the parental vector FU-CGW[21] by sequentially inserting matched ten-nucleotide barcodes into the PacI site distal to the HIV FLAP using the Quick Ligation Kit (New England Biolabs) and PCR amplification of the WPRE sequence and barcode with insertion into the ClaI sites proximal to the 3′ LTR by HiFi DNA Assembly (New England Biolabs). ORFs were cloned into the EcoRI site of the double-barcoded lentiviral vectors by HiFi DNA Assembly. To generate shRNA lentiviral vectors, the ubiquitin C promoter sequence was excised from the double-barcoded plasmid by digesting with PspXI and EcoRI. U6 promoter and shRNA cassettes were isolated by digesting pLKO.1 TRC shRNA clones with PspXI and EcoRI and were inserted into the digested double-barcoded plasmid using the Quick Ligation Kit. Individual lentiviruses were generated in arrayed format in 293T cells (CRL-11268, ATCC) by co-transfection of each double-barcoded lentiviral ORF or shRNA plasmid with the helper plasmids pVSV-G, pMDL and pRev using FuGENE HD Transfection Reagent (Promega). Lentiviral supernatants were collected 36 h after transfection, pooled and concentrated by ultracentrifugation in V-bottom polypropylene centrifuge tubes on a SW 32 Ti in an Optima XE 90 (Beckman Coulter) at $82,520g$ at 4 °C for 2 h. Supernatants were aspirated, and lentiviral pellets were resuspended in residual media and cryopreserved.

### shRNA screening

The top three to five shRNA sequences identified from The RNAi Consortium for each target gene were identified from the Genetic Perturbation Platform Web Portal at the Broad Institute. shRNA sequences were cloned into pLKO.1. pLKO.1-TRC control and pLKO.1-shRNA lentiviruses were generated and used to transduce 3T3 cells (gift from V. Vasioukhin, Fred Hutchinson Cancer Center). Seventy-two hours after lentiviral transduction, 3T3 cells were collected, and RNA was collected using an RNeasy Mini Kit (Qiagen). Reverse transcription of RNA was performed using SuperScript IV Reverse Transcriptase (Invitrogen) as per the manufacturer's instructions. qPCR was performed on a QuantStudio 6 using SYBR Green qPCR Master Mix (ThermoFisher Scientific), and primers specific to each target gene and *Ubc* as a control. All primers used for these studies are listed in Supplementary Table 3. Relative expression was calculated using ddCT analysis.

### Embryonic bladder mesenchyme and urogenital sinus mesenchyme preparation

All animal care and studies were performed in accordance with an approved Fred Hutchinson Cancer Center Institutional Animal Care and Use Committee (IACUC) protocol (PROTO000051048) and Comparative Medicine regulations. All animals were housed in an Association for Assessment and Accreditation of Laboratory Animal Care (AALAC)-accredited facility and subjected to a 12-h light/dark cycle with the temperature maintained between 18 °C and 24 °C and 40–60% humidity. UGSM was isolated and propagated as previously described[21]. E16 fetal bladders were also collected at the same time as the urogenital sinus and subjected to similar steps for preparation of EBLM. UGSM and EBLM were passaged less than five times before use in engraftment studies.

### Mouse bladder and prostate dissociation and organoid culture

Bladder and prostates from 8- to 12-week-old male C57BL/6 mice (The Jackson Laboratory) were dissected and mechanically and enzymatically dissociated as previously described[21]. Cells were stained with antibodies for FACS on a Sony SH800 Cell Sorter with collection of Lin⁻CD49fʰⁱᵍʰEpCAMʰⁱᵍʰ cells. Between $1 \times 10^4$ and $2 \times 10^4$ bladder urothelial and prostate epithelial cells were resuspended in a total of 15 μl of growth factor-reduced Matrigel (Corning) with or without

concentrated lentivirus and seeded as droplets in each 48-well tissue culture plate well. Cells were cultured as previously described[22]. Mouse bladder organoid culture media consisted of Advanced DMEM-F12, 10 mM HEPES, 2 mM GlutaMAX, B27 supplement, 1.25 mM N-acetylcysteine, 50 ng ml⁻¹ hEGF, 100 ng ml⁻¹ hNoggin and 500 ng ml⁻¹ hR-spondin, 200 nM A83-01 and 10 μM Y-27632. Mouse prostate organoid culture media consisted of mouse bladder organoid culture media with the addition of 1 nM dihydrotestosterone.

### Organoid transformation assay

After 5–7 days of culture, transduced mouse bladder urothelial or prostate epithelial organoids were liberated by dissociating the Matrigel matrix with 5 U ml⁻¹ dispase (STEMCELL Technologies). Organoids were washed with PBS and resuspended in ice-cold Matrigel with either $10^5$ EBLM or UGSM and subcutaneously injected into the flanks of 6- to 8-week-old male NSG (NOD-SCID-IL2Ry-null) mice (The Jackson Laboratory). For prostate epithelial transformation studies, mice were supplemented with testosterone through the subcutaneous implantation of 90-day release testosterone pellets (Innovative Research of America). Tumors were collected when they reached 1 cm in maximal diameter. The maximum tumor size permitted by the Fred Hutchinson Cancer Center IACUC is 2 cm in diameter, which was not exceeded during these studies.

### Copy number assay

DNA was extracted from organoids using a GeneJET Genomic DNA Purification Kit (ThermoFisher Scientific). Copy number analysis was performed by TaqMan Real-Time PCR Assay (ThermoFisher Scientific) using the TaqMan Copy Number Reference Assay, mouse, Tfrc (4458366) and EGFP TaqMan Copy Number Assay (Mr00660654_cn) on a QuantStudio 6. Genomic DNA extracted from the tails of transgenic C57BL/6 mice with one or two copies of GFP was used as a calibrator sample. GFP copy number was determined using ddCT analysis, where sample copy number = calibrator copy number $\times 2^{-\mathrm{ddCT}}$.

### Single-cell DNA amplicon sequencing library preparation and sequencing

A custom panel was designed for the Mission Bio Tapestri to amplify segments of ten mouse genes at two exons each, the 5′ and 3′ lentiviral barcodes and lentiviral GFP. Libraries were generated either from cryopreserved or freshly dissociated tumor cells using the Mission Bio Tapestri Single-cell DNA Custom Kit according to the manufacturer's recommendations. Single cells (3,000 to 4,000 cells per μl) were resuspended in Tapestri cell buffer and encapsulated using a Tapestri microfluidics cartridge, lysed and barcoded. Barcoded samples were subjected to targeted PCR amplification, and PCR products were removed from individual droplets, purified with KAPA Pure Beads (Roche Molecular Systems) and used as a template for PCR to incorporate Illumina P7 indices. PCR products were purified by KAPA Pure Beads and quantified by Qubit dsDNA High Sensitivity Assay (ThermoFisher Scientific). Sample quality was assessed by Agilent TapeStation analysis. Libraries were pooled and sequenced on an Illumina MiSeq or HiSeq 2500 with 150 bp paired-end reads in the Fred Hutchinson Cancer Center Genomics Shared Resource.

### Laser capture microdissection and DNA and RNA isolation for high-throughput sequencing

Sections 10 μm thick were cut from formalin-fixed paraffin-embedded (FFPE) tumor tissue blocks and mounted onto PEN Membrane Frame Slides (ThermoFisher Scientific). Sections were fixed with 95% ethanol for 1 min, stained with 3% cresyl violet and dehydrated through graded alcohols and xylene. Histology review and annotation were performed by a pathologist. Laser capture microdissection was performed on an Arcturus XT Laser Capture Microdissection System (ThermoFisher Scientific). Microdissected specimens were collected for DNA and

RNA extraction. DNA was extracted using a GeneRead DNA FFPE Kit (Qiagen), and RNA was extracted using an RNeasy FFPE Kit (Qiagen) according to the manufacturer's protocols. Two-step PCR for lentiviral barcode amplification and sequencing library adaptor ligation was performed. The first PCR reaction consisted of 2x KAPA HiFi HotStart ReadyMix, 100 nM of 1° FWD primer (5′- TCGTCGGCAGCGTCAGAT-GTGTATAAGAGACAGCAAAATTTTCGGGTT TATTACAGG-3′), 100 nM of 1° REV primer (5′- GTCTCGTGGGCTCGGAGATGTGTATAAGAGA CAGGCCGCTCGAGGACTATTAAG-3′) and 80 ng of genomic DNA. Thermal cycling conditions were 95 °C for 3 min; (95 °C for 30 s, 64 °C for 30 s, 72 °C for 30 s) × 25 cycles; 72 °C for 5 min; and hold at 4 °C. PCR cleanup was conducted using the Wizard SV Gel and PCR Clean-Up System (Promega), with elution in 30 μl of double distilled water. The second PCR reaction consisted of 2x KAPA HiFi HotStart ReadyMix, 140 nM of 2° i7 primer, 140 nM of 2° i5 primer and 5 μl of elution from the PCR cleanup of the 1° PCR. Thermal cycling conditions were 95 °C for 3 min, (95 °C for 30 s, 61 °C for 30 s, 72 °C for 30 s) × 8 cycles; 72 °C for 5 min; and hold at 4 °C. The sequences of 2° primers used to incorporate dual-indexed Illumina sequencing adaptors are displayed in Supplementary Table 4. PCR cleanup was conducted using the Wizard SV Gel and PCR Clean-Up System, with elution in 30 μl of double distilled water. Sample quality was assessed by Agilent TapeStation analysis. Sequencing was performed on an Illumina MiSeq or HiSeq 2500 instrument using 150 bp single-end reads. PhiX sequences were excluded from the sequencing reads by Bowtie 2 v2.4.4 (ref. [23]). Cutadapt v4.1 (ref. [24]) was used to trim the reads to the barcode region. Then the trimmed reads were aligned to custom DNA references containing all barcodes using Bowtie 2. Samtools v1.11 (ref. [25]) was used to extract read counts for each barcode. The RNA-seq libraries were prepared using a SMARTer Stranded Total RNA-Seq Kit v3 - Pico Input Mammalian (Takara Bio) and sequenced on an Illumina NovaSeq 6000 using a NovaSeq S4 flow cell with 100 bp paired-end reads by MedGenome. Sequencing reads were mapped to mouse genome reference GRCm39, and gene expression was quantified and normalized using the UC Santa Cruz Computational Genomics Lab Toil RNA-seq pipeline v4.1.2 (ref. [26]).

## Transcriptional subtype analysis and PCA projections

All computational analyses were carried out in RStudio v4.1.0. Mouse Ensembl genes were converted to Mouse Genome Informatics (MGI) gene symbols using the biomaRt package v2.24.1 (https://bioconductor.org/packages/release/bioc/html/biomaRt.html). MGI gene symbols were then converted to their human orthologs by referencing the mouse–human ortholog database available from The Jackson Laboratory (http://www.informatics.jax.org/downloads/reports/HOM_MouseHumanSequence.rpt). The human ortholog matrix was used for downstream analysis in transcriptional subtype analysis. FHBT samples were classified using the BASE47 subtype predictor gene list and the ConsensusMIBC package v1.1 (https://github.com/cit-bioinfo/consensusMIBC). Z-score means of genes and signature scores were calculated for each sample. Heatmaps of both the BASE47 and ConsensusMIBC results were generated using the pheatmap package v1.0.12 (https://www.rdocumentation.org/packages/pheatmap/versions/1.0.12/topics/pheatmap). For PCA analysis, the FPKM human ortholog matrix was normalized by $\log_2 + 1$ transformation before performing mean-centered PCA using the prcomp package v3.6.2 (https://www.rdocumentation.org/packages/stats/versions/3.6.2/topics/prcomp). Visualization of the PCA plot was performed using the factoextra package v1.0.7 (https://cran.r-project.org/web/packages/factoextra/index.html) and ggpubr package v0.6.0 (https://www.rdocumentation.org/packages/ggpubr/versions/0.6.0).

For PCA projections, RNA-seq count data from the FHBT, GSE220999 and TCGA-BLCA datasets were transformed to counts per million, normalized and batch corrected using ComBat-seq[27] to compare across each dataset using the DGEobj.utils package v1.0.6

(https://rdrr.io/cran/DGEobj.utils). PCA projection of the FHBT data onto the TCGA-BLCA space was done by first generating a PCA of the TCGA-BLCA samples from the common genes between the FHBT and GSE220999 data. A PCA for both the FHBT and GSE220999 samples was then scaled by the eigenvalues of the TCGA-BLCA using the base package v3.6.2 (https://www.rdocumentation.org/packages/base/versions/3.6.2/topics/scale). A plot was constructed overlaying the reference TCGA-BLCA samples with either FHBT or GSE220999 tumor projections using ggplot2 v3.4.1 (https://cran.r-project.org/web/packages/ggplot2/index.html). TCGA-BLCA samples were colored by their Consensus Molecular Classifier subtype. Differential gene expression analysis was performed pairwise between FHBT histologies using the DESeq2 package v1.38.3 (ref. [28]). P values were generated via the Wald test and P-adjusted using the Benjamini–Hochberg correction. Pre-ranked GSEA (Broad Institute) was conducted by inputting a ranked list of differentially expressed genes based on $\log_{10}$-transformed P values from the DESeq2 analysis for each pairwise comparison. Dot plots were generated by plotting the normalized enrichment score and log-transformed false discovery rate for each pre-ranked GSEA output using ggplot2.

## Single-cell lentiviral barcode enumeration and clonality analysis

Raw sequencing reads were trimmed to the amplicon regions using the awk command. Barcode sequences in the reads were filtered and extracted using UMI-tools v1.0.0 (ref. [29]). Processed reads were aligned to custom references containing all amplicon sequences using bwa-mem v0.7.17-r1188 (ref. [30]). Samtools was used to extract amplicon counts for each barcode. Mouse cells with no GFP amplicon counts were removed. Counts per cell were normalized to total counts for each barcode. A minimum threshold normalized count of 1% of total counts was used to define the presence of a barcode in a cell. The clonal architecture of cells was determined by enumerating all cells containing each distinct combination of barcodes.

## Immunohistochemistry

Tumor samples were formalin-fixed and paraffin-embedded, sectioned to a 5-μm thickness and placed on positively charged glass slides. For each tumor, slides were stained with a standard hematoxylin and eosin protocol. Immunohistochemical staining was performed according to an established protocol[31]. Stained slides were digitally scanned on a VENTANA DP 200 (Roche) and analyzed using QuPath 0.2.3 (ref. [32]).

## Antibodies

Antibodies used for FACS: Human/mouse/bovine integrin alpha 6/CD49f PE-conjugated antibody (FAB13501P, R&D Systems, 1:40); PE/Cyanine7 anti-mouse CD326 (Ep-CAM) antibody (118216, BioLegend, 1:40); CD31 (PECAM-1) monoclonal antibody (390), FITC (11-0311-82, eBioscience, 1:100); CD45 monoclonal antibody (30-F11), FITC (11-0451-85, eBioscience, 1:100); TER-119 monoclonal antibody (TER-119), FITC (11-5921-82, eBioscience, 1:100). Antibodies used for immunohistochemistry: Anti-wide spectrum Cytokeratin antibody (ab9377, Abcam, 1:100); rabbit monoclonal GFP antibody (clone D5.1, Cell Signaling, 1:100); rabbit polyclonal p63 antibody (12143-1-AP, Proteintech, 1:200); mouse monoclonal p53 antibody (clone 1C12, Cell Signaling, 1:500); rabbit monoclonal HOXB13 antibody (clone D7N8O, Cell Signaling, 1:50); rabbit polyclonal AR antibody (06-680, Millipore, 1:2,000); rabbit monoclonal GATA3 antibody (clone D13C9, Cell Signaling, 1:200); rabbit monoclonal CD44 antibody (clone E7K2Y, Cell Signaling, 1:100).

## Statistical analyses

Data analysis was performed on GraphPad Prism 9 (GraphPad Software). qPCR results were analyzed in Excel. Statistical significance was determined using the unpaired two-tailed Student's t-test. Results are

depicted as mean + s.d. unless stated otherwise. For all statistical tests, $P$ values of <0.05 were considered significant.

### Reporting summary

Further information on research design is available in the Nature Portfolio Reporting Summary linked to this article.

### Data availability

Sequencing data pertaining to this study are available from the Gene Expression Omnibus (GEO) as SuperSeries GSE229783. RNA-seq data from FHBT models are available from accession number GSE229780. Bulk DNA amplicon sequencing data from lentiviral library representation studies and from FHBT models are available from accession numbers GSE231542 and GSE229781, respectively. Single-cell DNA amplicon sequencing data related to determining the unique proviral copies per cell after lentiviral transduction across a range of MOIs are available from accession number GSE231543. Single-cell DNA amplicon sequencing data from FHBT models and enriched cells from the tumor model with prostate adenocarcinoma and focal pleomorphic giant cell carcinoma are available from accession number GSE229782.

### Code availability

The study did not use any custom code or software. All code or software used for all data processing and analysis has been described in the Methods section. No custom code was used to generate figures.

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

### Acknowledgements

We thank C. Morrissey, L. Xin and P. Nelson for critical discussion and review of this work. This work was supported by National Institutes of Health (NIH) grants DP2 CA271301 (J.K.L.), R01 CA276308 (A.C.H.), R37 CA230617 (A.C.H.), a Bladder Cancer Advocacy Network Research Innovator Award (J.K.L.), a Department of Defense Peer Reviewed Cancer Research Program Career Development Award W81XWH-19-1-0569 (J.K.L.), a Department of Defense Prostate Cancer Research Program Early Investigator Research Award W81XWH-20-1-0083 (S.L.) and a Department of Defense Peer Reviewed Cancer Research Program Horizon Award W81XWH-19-1-0658 (S.J.). We acknowledge support from the Seattle Translational Tumor Research Program in Bladder Cancer and the University of Washington Medicine Urethral Cancer Research Fund provided by donor D.L. Rich. This research was also supported by the Flow Cytometry, Experimental Histopathology and Genomics Shared Resources of the Fred Hutch/University of Washington Cancer Consortium funded by NIH grant P30 CA015704.

### Author contributions

S.L., A.W., H.S. and J.K.L. designed experiments. S.L., A.W., H.S., V.B., G.J. and S.J. performed the experiments. H.S. and G.J. analyzed the RNA-seq and bulk and single-cell DNA amplicon sequencing data. Q.W. provided statistical oversight. R.B.M., J.L.W., H.-M.L., A.C.H., B.M.F., M.C.H. and J.K.L. supervised the study. S.L. and J.K.L. wrote the manuscript with contributions from all authors.

### Competing interests

J.K.L. served on the Speaker's Bureau for Mission Bio. B.M.F. has consulting or advisory roles with QED Therapeutics, Boston Gene, Astrin Biosciences Merck, Immunomedics/Gilead, Guardant and Janssen and receives patent royalties from Immunomedics/Gilead and research support from Eli Lilly. All other authors declare no competing interests.

### Additional information

**Extended data** is available for this paper at https://doi.org/10.1038/s41588-024-01674-1.

**Correspondence and requests for materials** should be addressed to John K. Lee.

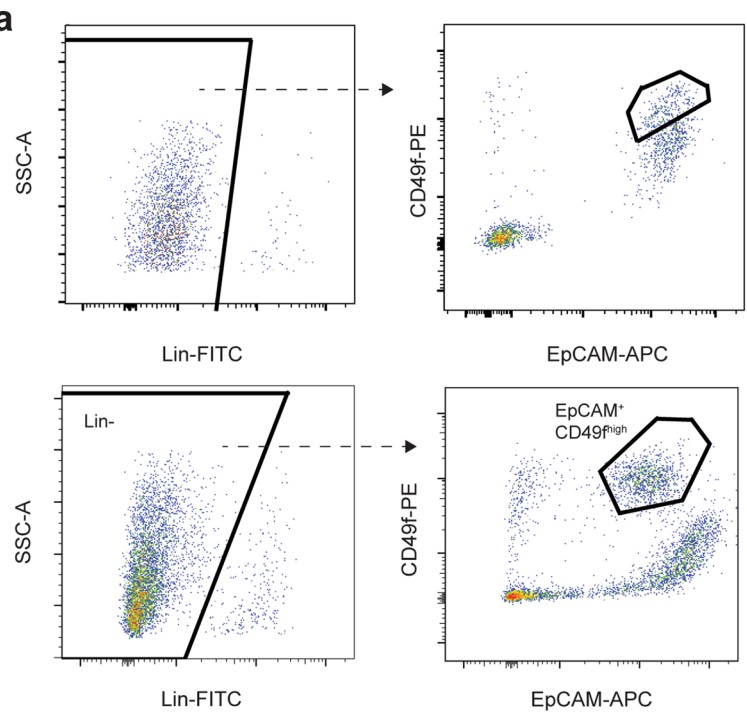

**a**

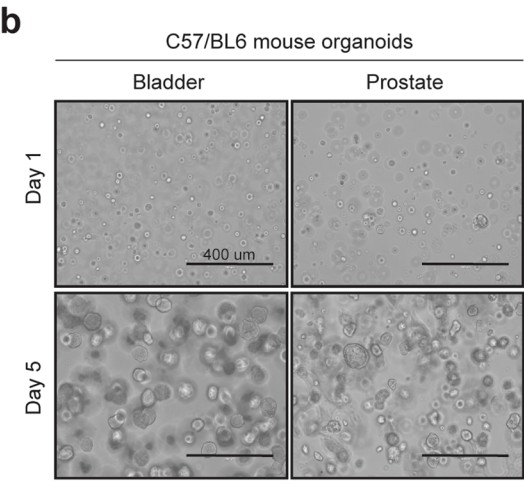

**b**

**c** Mission Bio Tapestri custom single-cell DNA amplicon sequencing panel design

| Amplicons | Source | Chr | Start | End | Bases |
|---|---|---|---|---|---|
| 5' BC | LV | | | | 10 |
| 3' BC | LV | | | | 10 |
| GFP | LV | | | | 100 |
| Trfc exon 5 | Ms | chr16 | 32616748 | 32616899 | 148 |
| Trfc exon 7 | Ms | chr16 | 32618215 | 32618333 | 115 |
| Actb exon 2 | Ms | chr5 | 142905565 | 142905684 | 120 |
| Actb exon 5 | Ms | chr5 | 142904071 | 142904250 | 176 |
| Gapdh exon 4 | Ms | chr6 | 125162850 | 125163024 | 175 |
| Gapdh exon 5 | Ms | chr6 | 125162506 | 125162669 | 147 |
| Hprt1 exon 3 | Ms | chrX | 53002100 | 53002255 | 156 |
| Hprt1 exon 4 | Ms | chrX | 53008657 | 53008716 | 60 |
| Pgk1 exon 3 | Ms | chrX | 106194360 | 106194504 | 145 |
| Pgk1 exon 5 | Ms | chrX | 106196843 | 106196944 | 102 |
| Sdha exon 9 | Ms | chr13 | 74332115 | 74332261 | 147 |
| Sdha exon 12 | Ms | chr13 | 74327249 | 74327349 | 101 |
| Ppia exon 3 | Ms | chr11 | 6418243 | 6418284 | 42 |
| Ppia exon 4 | Ms | chr11 | 6419159 | 6419202 | 44 |
| Rpl37 exon 2 | Ms | chr15 | 5117297 | 5117410 | 114 |
| Rpl37 exon 3 | Ms | chr15 | 5117626 | 5117696 | 71 |
| Eif3f exon 3 | Ms | chr7 | 108938038 | 108938110 | 73 |
| Eif3f exon 4 | Ms | chr7 | 108938409 | 108938531 | 123 |
| Eef2 exon 2 | Ms | chr10 | 81177736 | 81177866 | 127 |
| Eef2 exon 7 | Ms | chr10 | 81179606 | 81179784 | 175 |

**d**

| 3T3 Cell Population | Expected Frequency (%) | Tapestri Clonal Frequency (%) |
|---|---|---|
| Unlabeled | 28.13 | 28.59 |
| BC01+BC02+BC04+BC06 | 12.46 | 9.98 |
| BC02+BC04+BC06 | 1.41 | 6.12 |
| BC06+BC08 | 56.26 | 54.8 |
| BC05+BC07 | 0.14 | 0.51 |

**e**

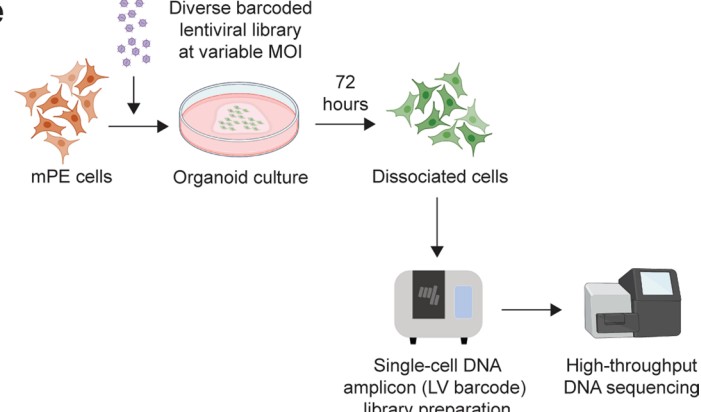

**Extended Data Fig. 1 | Isolation of mouse bladder urothelial and prostate epithelial cells for organoid culture and design/validation of a custom Mission Bio Tapestri single-cell DNA amplicon sequencing panel.**
(**a**) Representative flow cytometry plot for the isolation of mouse bladder urothelial (top) and prostate epithelial (bottom) from dissociated tissues based on a Lin$^-$ (CD45$^-$CD31$^-$Ter119$^-$) EpCAM$^+$CD49f$^{high}$ immunophenotype. (**b**) Images of organoid cultures of mouse bladder urothelial and prostate epithelial cells on day 1 and day 5 after seeding. (**c**) Table showing the amplicons represented in a custom Mission Bio Tapestri single-cell DNA amplicon sequencing panel.

(**d**) Table showing results of a validation study where a defined mixture of 3T3 cells with an unlabeled population and others labeled with combinations of lentiviruses encoding distinct barcodes were analyzed using the Mission Bio Tapestri single-cell DNA amplicon sequencing panel to determine clonality. ~2,000 cells were analyzed. (**e**) Overview of experiments with infection of mouse prostate epithelial (mPE) cells with a diverse barcoded lentiviral library in organoid culture across a range of multiplicity-of-infection (MOI) and quantification of viral copy number per cell across the population by single-cell amplicon sequencing. Created with BioRender.com.

**a** Genetic alterations associated with bladder cancer

| Gain-of-function | Loss-of-function |
|---|---|
| E2f3 | Kmt2d |
| Pparg | Kdm6a |
| Mdm2 | Arid1a |
| Ccnd1 | Kmt2c |
| Egfr | Ep300 |
| Pvrl4 | Stag2 |
| Ywhaz | Atm |
| Yap1 | Fat1 |
| Ccne1 | Sptan1 |
| Myc | Kmt2a |
| Zfp703 | Cdkn2a |
| Trp53 R245Q | Rb1 |
| Pik3ca E545K | Crebbp |
| Fgfr3 S243C | Ncor1 |
| Erbb3 V104L | Pten |
| Erbb2 S311Y | |

Genetic alterations associated with prostate cancer

| Gain-of-function | Loss-of-function |
|---|---|
| Ar | Pten |
| Erg | Ncor1 |
| Etv1 | Spen |
| Pik3ca E545K | Apc |
| Trp53 R245Q | Brca2 |
| Spop F133C | Atm |
| Foxa1 R261C | Cdk12 |
| Ctnnb1 D32A | Cdkn1b |
| Myc | Kmt2c |
| | Kmt2d |
| | Kdm6a |
| | Chd1 |
| | Ttn |
| | Zfhx3 |

**b**

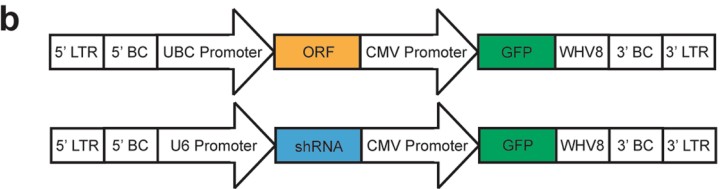

**c**

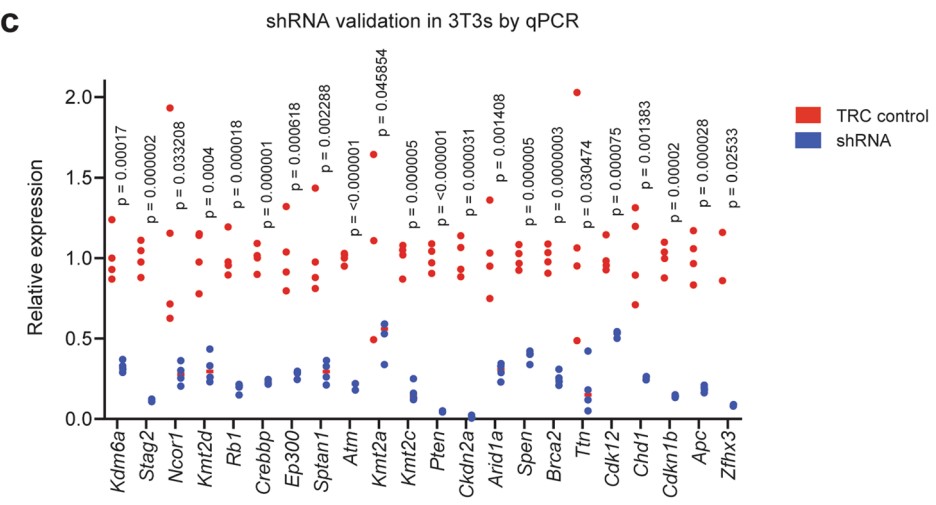

**d**

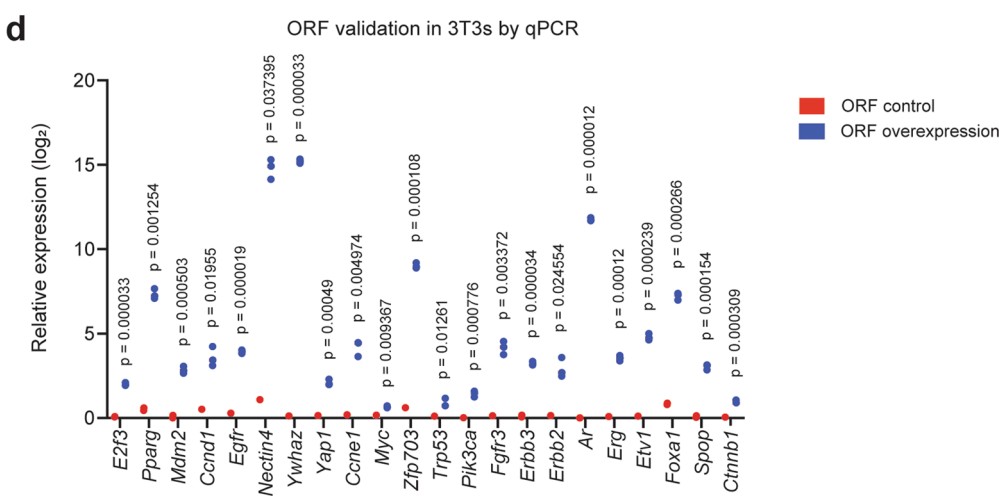

Extended Data Fig. 2 | See next page for caption.

**Extended Data Fig. 2 | Recurrent genetic alterations associated with bladder and prostate cancer encoded in barcoded lentiviral libraries.** (**a**) Tables showing gain-of-function and loss-of-function genetic alterations associated with bladder and prostate cancer selected for representation in cancer-specific barcoded lentiviral libraries. (**b**) Schematics of barcoded lentiviral vectors expressing open reading frames (ORF) or short-hairpin RNA (shRNA). LTR = long terminal repeat; BC = barcode; UBC = Ubiquitin C; CMV = cytomegalovirus; GFP = green fluorescent protein; WHV8 = Woodchuck hepatitis virus 8 post-transcriptional regulatory element. (**c**) Plot showing relative expression of target genes as determined by quantitative polymerase chain reaction (qPCR) in 3T3 cells 72 hours after lentiviral transduction with pLKO.1-TRC control or pLKO.1 expressing select shRNA previously screened and selected for inclusion in the barcoded lentiviral libraries based on the extent of gene knockdown. qPCR reactions were performed on four biologically independent replicates. Statistical analysis was performed by two-tailed, unpaired $t$-test with p-values shown. (**d**) Plot showing relative overexpression of gain-of-function genes as determined by qPCR in 3T3 cells 72 hours after lentiviral transduction with barcoded ORF vectors. qPCR reactions were performed on three biologically independent replicates. Statistical analysis was performed by two-tailed, unpaired $t$-test with p-values shown.

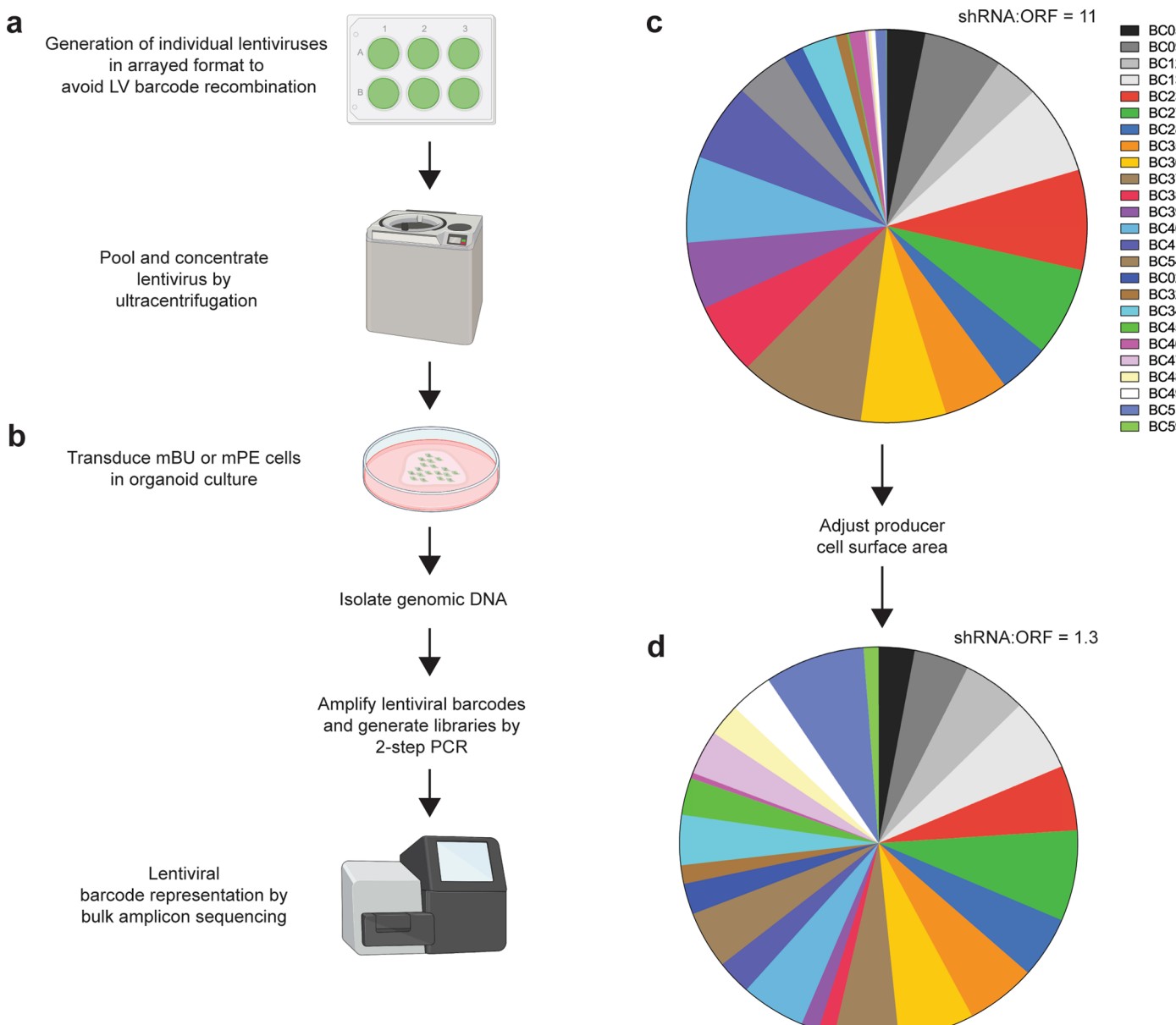

**Extended Data Fig. 3 | Generation of barcoded lentiviral libraries and normalization of library representation.** Schema showing the (**a**) generation of individual lentiviruses from the library in arrayed format with subsequent pooling and concentration by ultracentrifugation and (**b**) transduction of respective mouse bladder urothelial (mBU) or prostate epithelial (mPE) cells in organoid culture with concentrated lentiviral libraries to determine lentiviral barcode representation by bulk amplicon sequencing of genomic DNA. Created with BioRender.com. (**c**) Representative distribution of barcoded lentiviruses within a library with skewed enrichment of shRNA relative to ORF lentiviruses. (**d**) Representative distribution of barcoded lentiviruses within a library after applying information from **c** to adjust producer cell surface area in **a** for the generation of the lentiviral library.

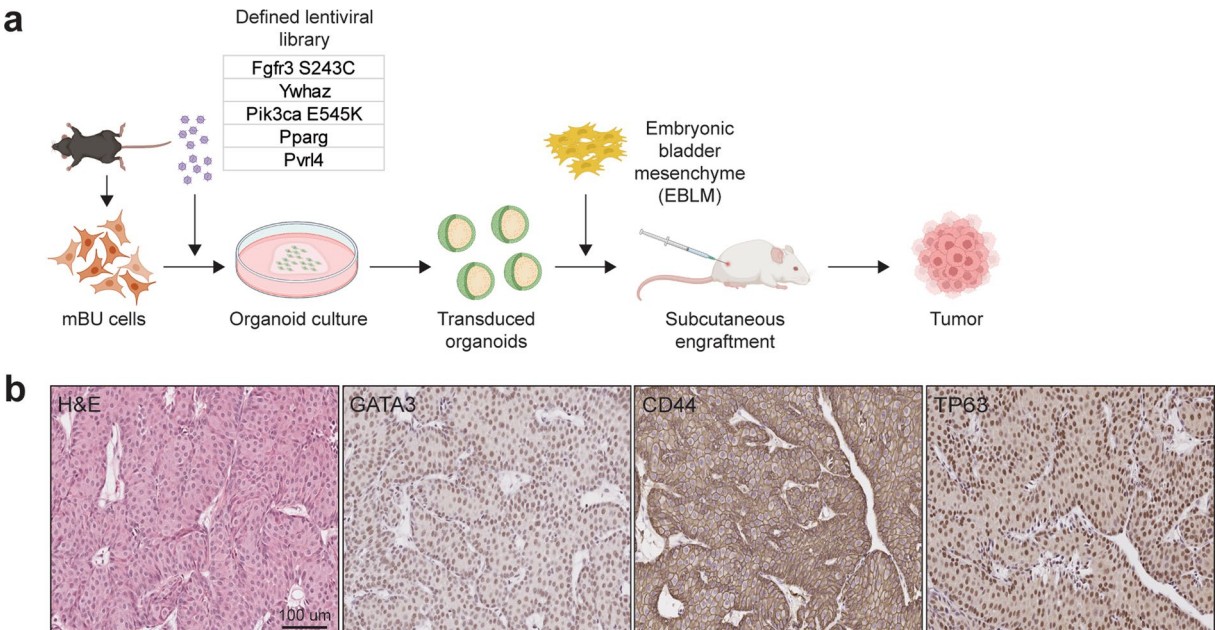

**Extended Data Fig. 4 | Active mutant Fgfr3 S243C cooperates with other oncogenic factors in mouse bladder urothelial cells to drive papillary urothelial carcinoma with inverted growth pattern. (a)** Scheme of the mBU organoid transformation assay using a defined lentiviral library to confirm functional genotype-phenotype associations. Created with BioRender.com. **(b)** High-magnification images of H&E- and IHC-stained sections of a resultant tumor of the experiment in **a** with histologic features consistent with papillary urothelial carcinoma with inverted growth pattern.

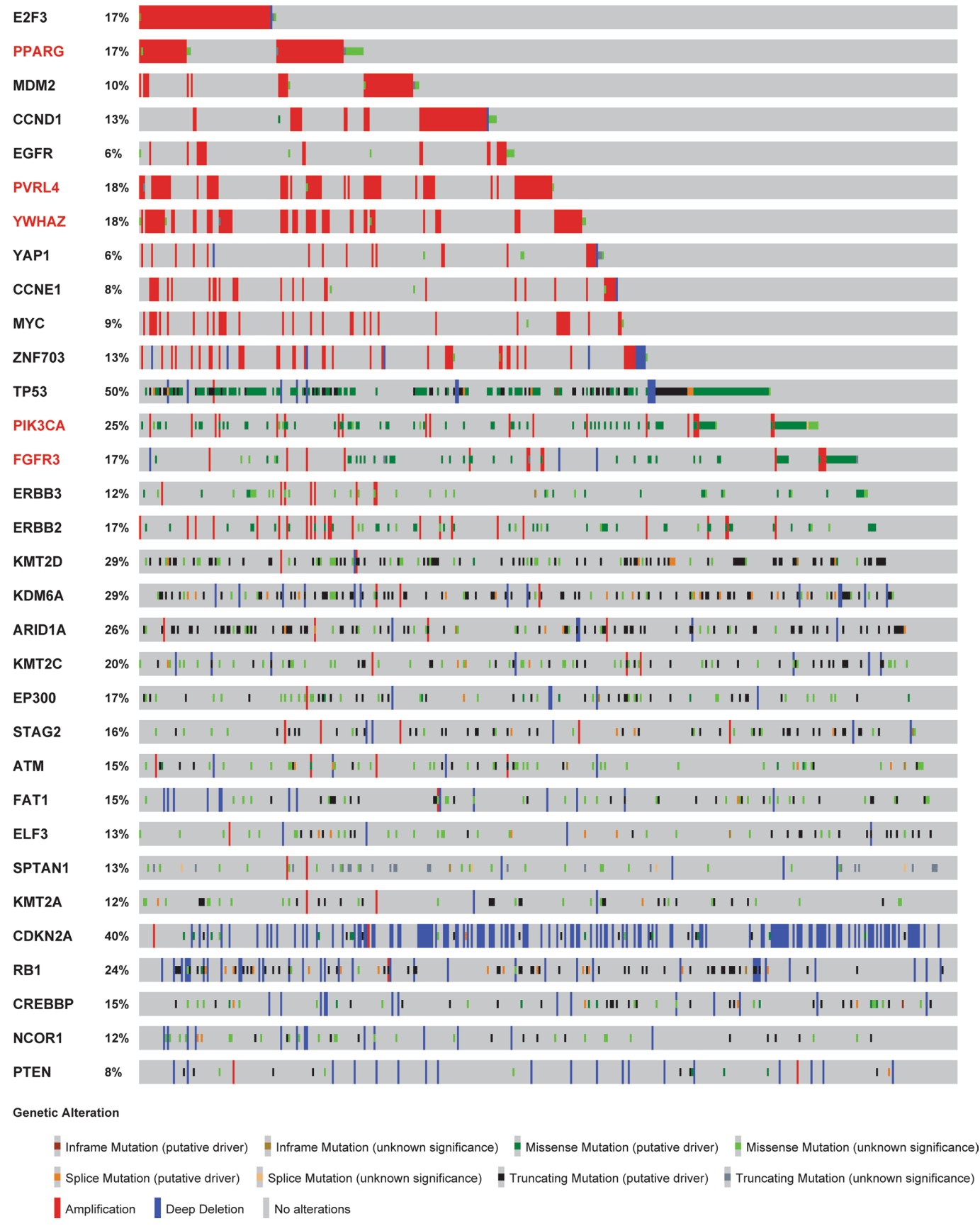

**Extended Data Fig. 5 | Frequencies of gene alterations represented in the bladder urothelial LV pool in human muscle-invasive bladder cancer.** Oncoprint from cBioPortal analysis of human muscle-invasive bladder cancers from The Cancer Genome Atlas bladder cancer (TCGA-BLCA) cohort showing select genes for which gain- or loss-of-function events were incorporated into the bladder urothelial LV pool.

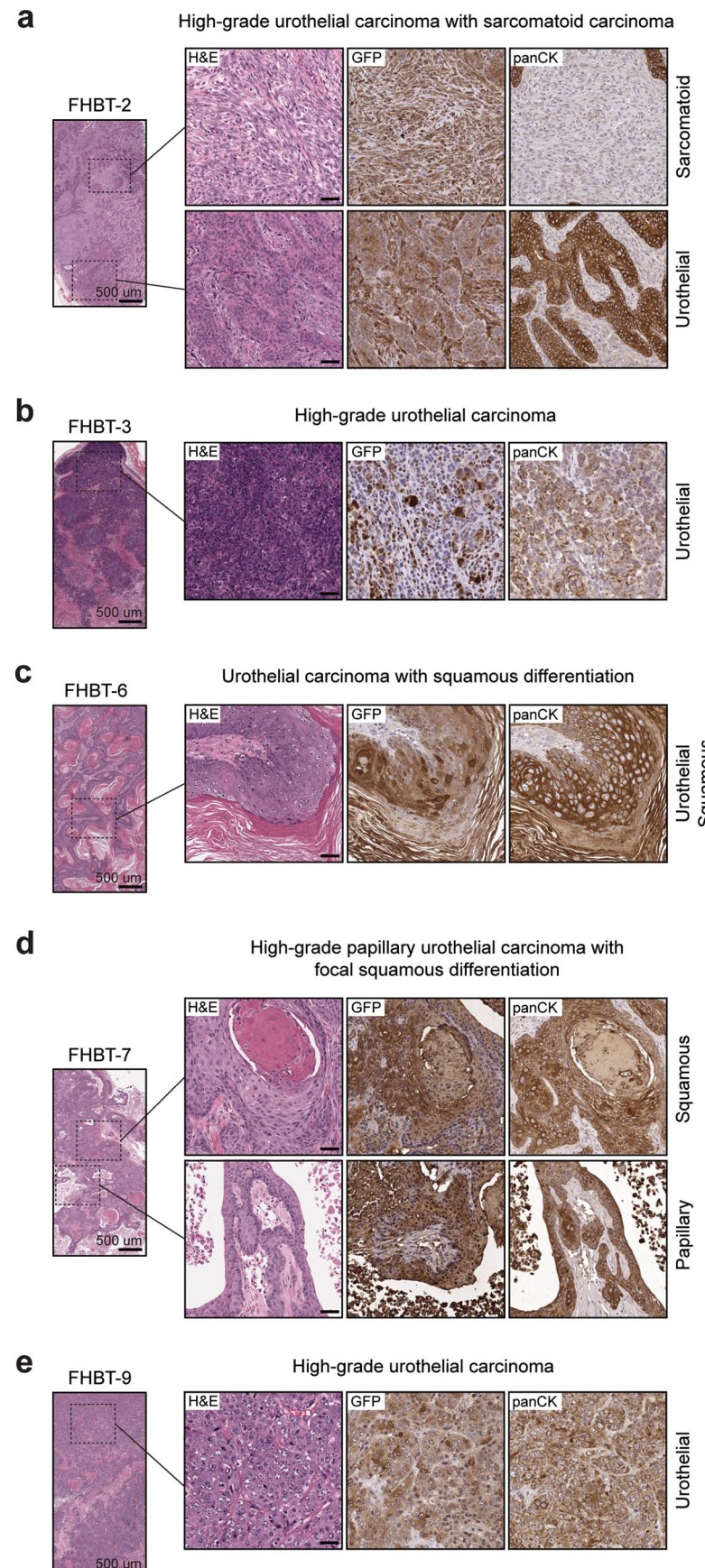

**Extended Data Fig. 6 | FHBT models demonstrate diverse cancer histologies. (a–e)** Low- and high-magnification images of H&E-stained sections and high-magnification images of IHC-stained sections for GFP and pan-cytokeratin (panCK) expression depicting characteristic histologies. Scale bars = 50 µm.

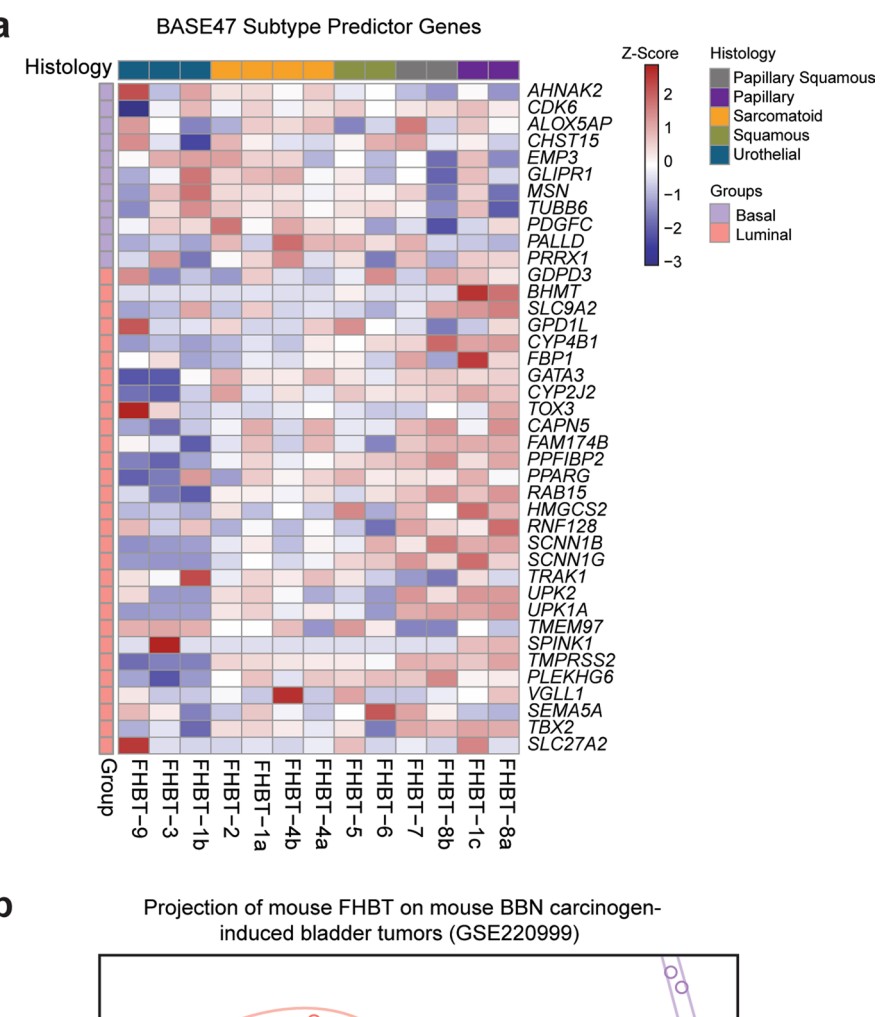

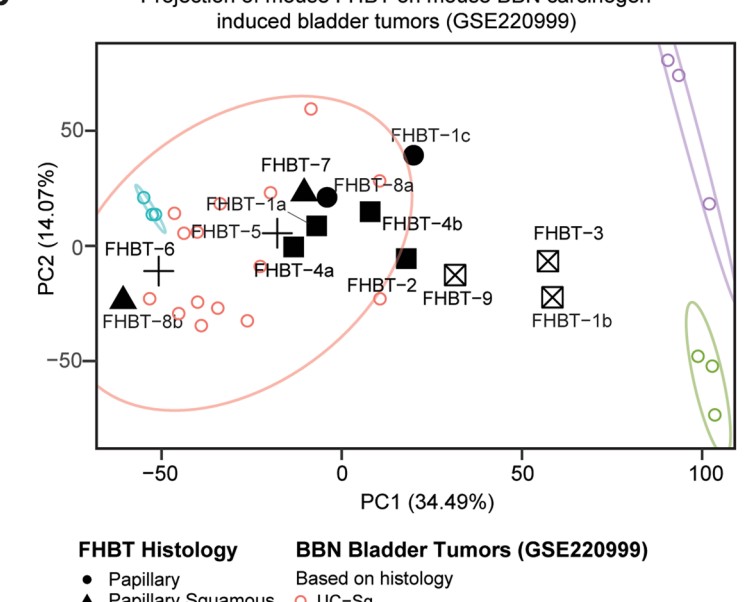

**Extended Data Fig. 7 | Phenotypic diversity and relevance of FHBT models.** (**a**) Heatmap showing the histologies of the Fred Hutch Bladder Tumor (FHBT) series relative to expressions of genes that constitute basal and luminal signatures for the BASE47 subtype predictor. (**b**) Principle component analysis (PCA) projection plot of FHBT samples over BBN carcinogen-induced mouse bladder tumors color-coded based on histology or histology and Consensus Molecular Classification (UC = urothelial carcinoma, Sq = squamous, Src = sarcomatoid, NE = neuroendocrine) with 90% confidence ellipses shown.

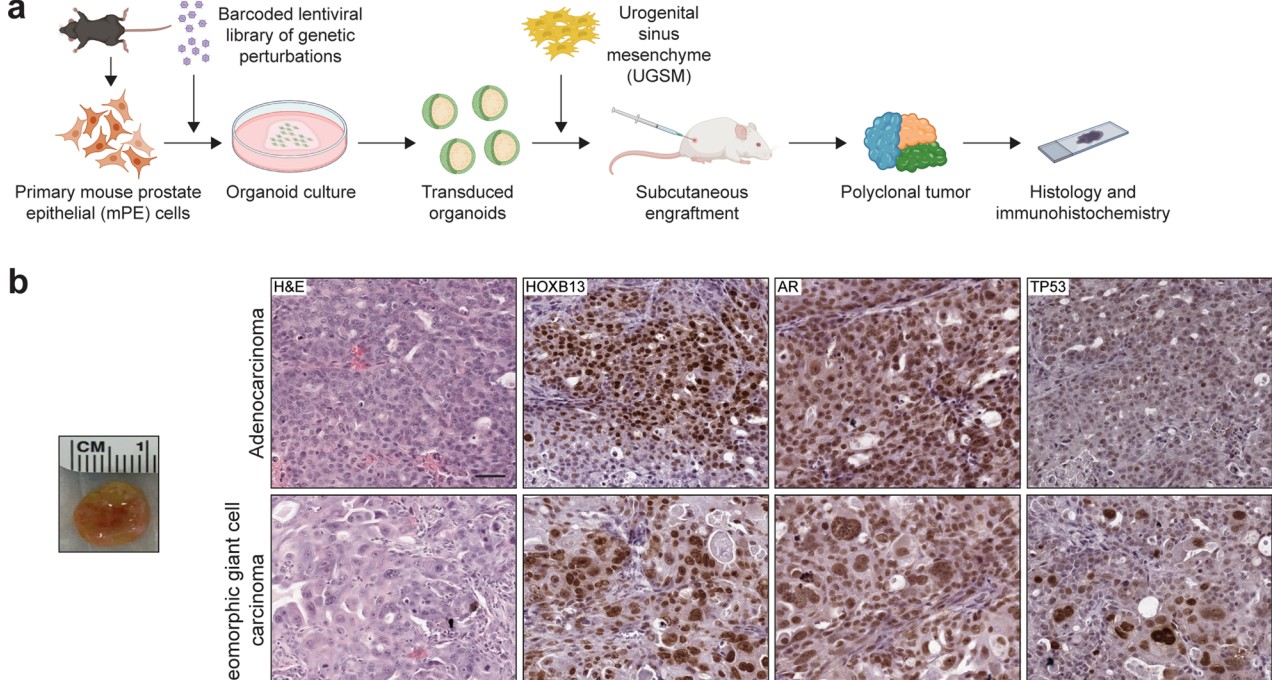

**a**

Primary mouse prostate epithelial (mPE) cells — Organoid culture — Transduced organoids — Subcutaneous engraftment — Polyclonal tumor — Histology and immunohistochemistry

Barcoded lentiviral library of genetic perturbations

Urogenital sinus mesenchyme (UGSM)

**b**

Adenocarcinoma / Pleomorphic giant cell carcinoma — H&E | HOXB13 | AR | TP53

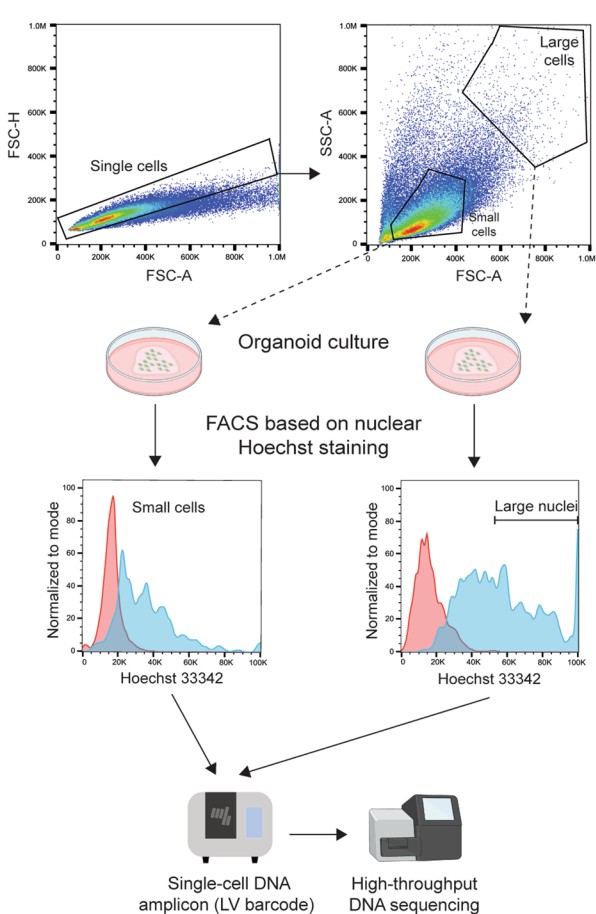

**c**

FACS of dissociated mixed prostate adenocarcinoma and pleomorphic giant cell carcinoma tumor based on cell size

Single cells / Large cells / Small cells

Organoid culture

FACS based on nuclear Hoechst staining

Small cells / Large nuclei

Single-cell DNA amplicon (LV barcode) library preparation — High-throughput DNA sequencing

**d**

Single-cell clonality analysis of small cells/nuclei

| Genes encoded by LV | Clonal frequency (%) |
|---|---|
| Etv1 | 34.3 |
| Spop F133C, Etv1 | 26.1 |
| shZfhx3 | 5.0 |
| Spop F133C, shZfhx3 | 2.9 |
| Myc, shZfhx3 | 2.6 |
| Myc, Spop F133C, shZfhx3 | 2.2 |
| Etv1, shZfhx3 | 1.9 |
| Myc, Etv1 | 1.9 |
| Spop F133C, Etv1, shZfhx3 | 1.6 |
| Trp53 R245Q, Etv1, shZfhx3 | 1.5 |

Single-cell clonality analysis of large cells/nuclei

| Genes encoded by LV | Clonal frequency (%) |
|---|---|
| shKmt2c, shTtn, Etv1 | 11.8 |
| shKmt2c, shTtn, Spop F133C, Etv1 | 5.5 |
| shKmt2c, shTtn, Erg, Etv1 | 5.1 |
| Myc, Etv1 | 4.7 |
| Spop F133C, Etv1 | 4.5 |
| shKmt2c, Etv1 | 4.1 |
| shKmt2c, shTtn, Spop F133C, Erg, Etv1 | 2.7 |
| shKmt2c, Spop F133C, Etv1 | 2.7 |
| shKmt2c, shTtn, Myc, Etv1 | 2.7 |
| Myc, Spop F133C, Etv1 | 2.3 |
| shKmt2c, Spop F133C, Erg, Etv1 | 2.1 |
| shKmt2c, Myc, Etv1 | 1.9 |
| shKmt2c, Pik3ca E545K, Spop F133C, Erg, Etv1 | 1.9 |
| shKmt2c, Erg, Etv1 | 1.8 |
| shKmt2c, Myc, Spop F133C, Etv1 | 1.7 |
| shKmt2c, Pik3ca E545K, shTtn, Erg, Etv1 | 1.7 |
| shKmt2c, Pik3ca E545K, shTtn, Etv1 | 1.5 |

**Extended Data Fig. 8 | See next page for caption.**

**Extended Data Fig. 8 | Association of adenocarcinoma with polymorphic giant cell carcinoma of the prostate with perturbation of Kmt2c.**
(**a**) Scheme of the mouse prostate epithelial (mPE) organoid transformation assay to uncover functional genotype-phenotype associations in prostate cancer. Created with BioRender.com. (**b**) *Left*, Gross image of a tumor arising from mPE transformed with a prostate epithelial lentiviral pool (PE-LVp). *Right*, high-magnification images of H&E- and IHC-stained sections of regions with high-grade adenocarcinoma and pleomorphic giant cell carcinoma. Scale bar = 50 μm. (**c**) Overview of the experimental approach to enrich for prostate adenocarcinoma and pleomorphic giant cell carcinoma based on cell size and nuclear DNA content followed by single-cell lentiviral barcode enumeration. (**d**) Tables showing single-cell clonality analysis of: *Top*, tumor cells enriched for 'small cells/nuclei.' *Bottom*, tumor cells enriched for 'large cells/nuclei.' Highlighted in red is sh*Kmt2c* based on the enumeration of the associated lentiviral barcode. Created with BioRender.com.

# Reporting Summary

## Statistics

For all statistical analyses, confirm that the following items are present in the figure legend, table legend, main text, or Methods section.

| n/a | Confirmed | |
|---|---|---|
| ☐ | ☒ | The exact sample size (*n*) for each experimental group/condition, given as a discrete number and unit of measurement |
| ☐ | ☒ | A statement on whether measurements were taken from distinct samples or whether the same sample was measured repeatedly |
| ☐ | ☒ | The statistical test(s) used AND whether they are one- or two-sided *Only common tests should be described solely by name; describe more complex techniques in the Methods section.* |
| ☒ | ☐ | A description of all covariates tested |
| ☐ | ☒ | A description of any assumptions or corrections, such as tests of normality and adjustment for multiple comparisons |
| ☐ | ☒ | A full description of the statistical parameters including central tendency (e.g. means) or other basic estimates (e.g. regression coefficient) AND variation (e.g. standard deviation) or associated estimates of uncertainty (e.g. confidence intervals) |
| ☐ | ☒ | For null hypothesis testing, the test statistic (e.g. *F*, *t*, *r*) with confidence intervals, effect sizes, degrees of freedom and *P* value noted *Give P values as exact values whenever suitable.* |
| ☒ | ☐ | For Bayesian analysis, information on the choice of priors and Markov chain Monte Carlo settings |
| ☐ | ☒ | For hierarchical and complex designs, identification of the appropriate level for tests and full reporting of outcomes |
| ☒ | ☐ | Estimates of effect sizes (e.g. Cohen's *d*, Pearson's *r*), indicating how they were calculated |

*Our web collection on statistics for biologists contains articles on many of the points above.*

## Software and code

Policy information about availability of computer code

| Data collection | The single-cell or bulk DNA amplicon sequencing data collected for this manuscript was generated using a custom panel designed for the Mission Bio Tapestri to amplify segments of ten mouse genes at two exons each, the 5' and 3' lentiviral barcodes, and lentiviral GFP. Amplicon DNA libraries generated using Tapestri were sequenced on an Illumina MiSeq or HiSeq 2500 with 150 bp paired-end reads in the Fred Hutchinson Cancer Center Genomics Shared Resource. The bulk RNA sequencing libraries were prepared using a SMARTer Stranded Total RNA-Seq Kit v3 - Pico Input Mammalian (Takara Bio) and sequenced on an Illumina NovaSeq 6000 using a NovaSeq S4 flow cell with 100 bp paired end reads by MedGenome, Inc. |
|---|---|
| Data analysis | Data and statistical analysis were performed using Microsoft Excel Office vl6.65 and Graph Pad Prism v9.4.l. Standard statistical tests were used to analyze biological data including Student's t test,W ilcoxon rank-sum test,F isher's exact test,t wo-way ANOVA with post hoc Tukey's or Sidak's multiple comparisons. Results for immunohistochemical (IHC) analysis were plotted using QuPath 0.2.3. Results from flow cytometric analysis were acquired using BD FACSCanto and Sony SH800 Cell Sorter instruments. Flow data were analyzed using FlowJo 10.8.0. Bulk RNA-seq and single-cell or bulk DNA amplicon sequencing data were analyzed on a Linux workstation using R v4.l.0. The sequencing data reported in the manuscript was processed using the following tools or packages: RStudio v4.1.0,biomaRt package v2.24.1,the ConsensusMIBC package v1.1,pheatmap package v1.0.12,prcomp package v3.6.2,factoextra package v1.0.7,ggpubr package v0.6.0,D GEobj.utils package v1.0.6,scale: Scaling and Centering of Matrix-like Objects base package v3.6.2,ggplot2 v3.4.1,DESeq2 package v1.38.3,Gene Set Enrichment Analysis 4.3.1,samtools v1.11,BWA-mem v0.7.17-r1188,Bowtie2 v2.4.4,Cutadapt v4.1,UMI-tools vl.0.0,p ython 3.7,toil-rnaseq v4.1.2,sva (ComBat-seq) v3.36.0 ) |

For functional experiments, each was repeated at least three times independently and results were expressed as mean± SD or mean ± SEM.

For manuscripts utilizing custom algorithms or software that are central to the research but not yet described in published literature, software must be made available to editors and reviewers. We strongly encourage code deposition in a community repository (e.g. GitHub). See the Nature Portfolio guidelines for submitting code & software for further information.

## Data

Policy information about availability of data

All manuscripts must include a data availability statement. This statement should provide the following information, where applicable:
- Accession codes, unique identifiers, or web links for publicly available datasets
- A description of any restrictions on data availability
- For clinical datasets or third party data, please ensure that the statement adheres to our policy

Sequencing data pertaining to this study is available from Gene Expression Omnibus (GEO) as SuperSeries GSE229783. RNA-seq data from FHBT models is available from accession number GSE229780. Bulk DNA amplicon sequencing data from lentiviral library representation stud-ies and from FHBT models are available from accession numbers GSE231542 and GSE229781, respectively. Single-cell DNA amplicon sequencing data related to determining the unique proviral copies per cell after lentiviral transduction across a range of MOIs is available from accession number GSE231543. Single-cell DNA amplicon sequencing data from FHBT models and enriched cells from the tumor model with prostate adenocarcinoma and focal pleomorphic giant cell carcinoma is available from accession number GSE229782.

## Research involving human participants, their data, or biological material

Policy information about studies with human participants or human data. See also policy information about sex, gender (identity/presentation), and sexual orientation and race, ethnicity and racism.

| | |
|---|---|
| Reporting on sex and gender | NA |
| Reporting on race, ethnicity, or other socially relevant groupings | NA |
| Population characteristics | NA |
| Recruitment | NA |
| Ethics oversight | NA |

Note that full information on the approval of the study protocol must also be provided in the manuscript.

# Field-specific reporting

Please select the one below that is the best fit for your research. If you are not sure, read the appropriate sections before making your selection.

☒ Life sciences          ☐ Behavioural & social sciences          ☐ Ecological, evolutionary & environmental sciences

For a reference copy of the document with all sections, see nature.com/documents/nr-reporting-summary-flat.pdf

# Life sciences study design

All studies must disclose on these points even when the disclosure is negative.

| | |
|---|---|
| Sample size | In vitro experiments were performed using three independent replicates and each experiment was repeated at least three times. Sample sizes for in vitro experiments were either based on ensuring sufficient statistical power or based on the standard in the field. For in vivo studies, transduced mouse bladder or prostate cells were subcutaneously injected in 5-6 mice and those which formed tumors were collected and screened by histology. Sample size was based on prior experience with dissociated-cell tissue recombination/transplantation assays and inherent variability due to technical complexity and pilot studies of the frequency of transformation of prostate (30-40%) and bladder (70-80%) epithelial cell grafts using the methodology (see Extended Data Table 1). 5-6 grafts from each transformation study ensured the generation of tumors from at least one graft for each experiment. |
| Data exclusions | No data was excluded from the analyses. |
| Replication | Experiments have been repeated multiple times using different independent biological samples with similar experimental conditions or otherwise mentioned in the figure legends, main text or methods. |
| Randomization | Randomization was not applicable as there was no pre-specified comparison of interventions. |
| Blinding | Blinding was not applied to the study as a therapeutic intervention was not investigated. |

# Reporting for specific materials, systems and methods

We require information from authors about some types of materials, experimental systems and methods used in many studies. Here, indicate whether each material, system or method listed is relevant to your study. If you are not sure if a list item applies to your research, read the appropriate section before selecting a response.

## Materials & experimental systems

| n/a | Involved in the study |
|---|---|
| ☐ | ☒ Antibodies |
| ☐ | ☒ Eukaryotic cell lines |
| ☒ | ☐ Palaeontology and archaeology |
| ☐ | ☒ Animals and other organisms |
| ☒ | ☐ Clinical data |
| ☒ | ☐ Dual use research of concern |
| ☒ | ☐ Plants |

## Methods

| n/a | Involved in the study |
|---|---|
| ☒ | ☐ ChIP-seq |
| ☐ | ☒ Flow cytometry |
| ☒ | ☐ MRI-based neuroimaging |

## Antibodies

**Antibodies used**

Antibodies used in this study for FACS:
Human/mouse/bovine integrin alpha 6/CD49f PE-conjugated antibody (FAB13501P, R&D Systems, 1:40);
PE/Cyanine 7 anti-mouse CD325 (Ep-CAM) antibody (118216, Biolegend, 1:40);
CD31 (PECAM-1) monoclonal antibody (390), FITC (11-0311-82,eBioscience, 1:100);
CD45 monoclonal antibody (30-Fll), FITC (11-0451-85, eBioscience, 1:100);
TER-119 monoclonal antibody (TER-119), FITC (11-5921-82, eBioscience, 1:100).

Antibodies used for immunohistochemistry:
Rabbit polyclonal panCK (ab9377, Abcam, 1:100);
Rabbit monoclonal GFP antibody (clone D5.1, Cell Signaling, 1:100);
Rabbit polyclonal p63 antibody (12143-1-AP, Proteintech, 1:200);
Mouse monoclonal p53 antibody (clone 1C12, Cell Signaling, 1:500);
Rabbit monoclonal HOXB13 antibody (clone D7N8O, Cell Signaling, 1:50);
Rabbit polyclonal AR antibody (06-680, Millipore, 1:2,000);
Rabbit monoclonal GATA3 antibody (clone D13C9, Cell Signaling, 1:200);
Rabbit monoclonal CD44 antibody (clone E7K2Y, Cell Signaling, 1:100).

**Validation**

For each antibody, the validation statement has been taken from the manufacturer's website or data sheet and detailed as follows:

Human/mouse/bovine integrin alpha 6/CD49f PE-conjugated antibody (FAB13501P, R&D Systems) - the antibody is validated to detect human, mouse, and bovine Integrin alpha 6/CD49f. Recognizes an epitope in the extracellular domain of the Integrin alpha 6 subunit.

PE/Cyanine 7 anti-mouse CD325 (Ep-CAM) antibody (clone- G8.8, 118216, BioLegend)- validated to be used by flow and is cited in multiple publication can be seen on this website: https://www.biolegend.com/fr-lu/products/pe-cyanine7-anti-mouse-cd326-ep-cam-antibody-5303.

CD31 (PECAM-1) monoclonal antibody (390), FITC (11-0311-82,eBioscience) - validated by staining in more than 40 publications found on this website: https://www.thermofisher.com/antibody/product/CD31-PECAM-1-Antibody-clone-390-Monoclonal/11-0311-82.

CD45 monoclonal anitbody (30-F11), FITC (11-0451-85, eBioscience) - the 30-F11 antibody has been tested by flow cytometric analysis of mouse bone marrow cells. https://www.thermofisher.com/antibody/product/CD45-Antibody-clone-30-F11-Monoclonal/11-0451-82

TER-119 monoclonal antibody (TER-119), FITC (11-5921-82, eBioscience) - has been tested by flow cytometric analysis of mouse bone marrow cells. https://www.thermofisher.com/antibody/product/TER-119-Antibody-clone-TER-119-Monoclonal/17-5921-82

Rabbit polyclonal panCK (ab9377, Abcam, 1:100) - suitable for: IHC-P, ICC, ICC/IF, Flow Cyt, WB, IHC-Fr. https://www.abcam.com/products/primary-antibodies/wide-spectrum-cytokeratin-antibody-ab9377.html

Rabbit monoclonal GFP antibody (clone D5.1, Cell Signaling, 1:100) - validated for WB and IHC. https://www.cellsignal.com/products/primary-antibodies/gfp-d5-1-rabbit-mab/2956

Rabbit polyclonal p63 antibody (12143-1-AP, Proteintech, 1:200) - validated for WB, IHC, IP and IF. https://www.ptglab.com/products/TP63-Antibody-12143-1-AP.htm

Mouse monoclonal p53 antibody (clone 1C12, Cell Signaling, 1:500) - validated for WB, IHC, flow, and ChIP. https://www.cellsignal.com/products/primary-antibodies/p53-1c12-mouse-mab/2524

Rabbit monoclonal HOXB13 anti-body (clone D7N8O, Cell Signaling, 1:50) - Validated for WB, IP, and IHC. https://www.cellsignal.com/products/primary-antibodies/hoxb13-d7n8o-rabbit-mab/90944

Rabbit polyclonal AR antibody (06-680, Millipore, 1:2,000) - validated for WB and IHC. https://www.emdmillipore.com/US/en/product/Anti-Androgen-Receptor-Antibody,MM_NF-06-680

Rabbit monoclonal GATA3 antibody (clone D13C9, Cell Signaling, 1:200) - validated for WB, IHC, ChIP, IF and flow. https://www.cellsignal.com/products/primary-antibodies/gata-3-d13c9-xp-rabbit-mab/5852

Rabbit monoclonal CD44 antibody (clone E7K2Y, Cell Signaling, 1:100) - validated for WB and IHC. https://www.cellsignal.com/products/primary-antibodies/cd44-e7k2y-xp-rabbit-mab/37259

# Eukaryotic cell lines

Policy information about cell lines and Sex and Gender in Research

| Cell line source(s) | HEK293T (CRL-3216) were obtained from the American Type Culture Collection and were cultured in DMEM medium supplemented with 10% FBS, 100 U/mL penicillin and 100 µg/mL streptomycin, and 4 mmol/L GlutaMAX (Thermo Fisher). |
|---|---|
| Authentication | Cell line authentication was done via short tandem repeat (STR) profiling at the IDEXX BioAnalytics, 4011 Discovery Drive, Columbia, MO 65201. |
| Mycoplasma contamination | All cell lines routinely tested negative for Mycoplasma contamination. |
| Commonly misidentified lines (See ICLAC register) | No misidentified cell lines were used in this study. |

# Animals and other research organisms

Policy information about studies involving animals; ARRIVE guidelines recommended for reporting animal research, and Sex and Gender in Research

| Laboratory animals | For studies using immunocompromised mice, six- to eight-week-old male NSG (NOD-SCID-IL2Rγ-null) mice were obtained from The Jackson Laboratory and were 2-4 months old when used for the studies.<br>Eight- to twelve-week-old male wild-type C57Bl/6 (C57Bl/6J) mice were obtained from The Jackson Laboratory and were 2-4 months old when used for the studies. |
|---|---|
| Wild animals | The study did not involve wild animals. |
| Reporting on sex | Male mice were used for all the experiments. |
| Field-collected samples | The study did not include field-collected samples. |
| Ethics oversight | All animal care and studies were performed in accordance with an approved Fred Hutchinson Cancer Center Institutional Animal Care and Use Committee protocol (PROTO000051048) and Comparative Medicine regulations. |

Note that full information on the approval of the study protocol must also be provided in the manuscript.

# Flow Cytometry

## Plots

Confirm that:

☒ The axis labels state the marker and fluorochrome used (e.g. CD4-FITC).

☒ The axis scales are clearly visible. Include numbers along axes only for bottom left plot of group (a 'group' is an analysis of identical markers).

☒ All plots are contour plots with outliers or pseudocolor plots.

☒ A numerical value for number of cells or percentage (with statistics) is provided.

## Methodology

| Sample preparation | Bladder and prostates from eight- to twelve-week-old male C57BL/6 mice were dissociated into single cells and were stained with antibodies for fluorescence-activated cell sorting on a Sony SH800 Cell Sorter. Bladder urothelial and prostate epithelial cells were sorted and collected based on a Lin(-) CD49f(high) EpCAM(high) immunophenotype. For prostate polymorphic giant cells were analyzed based on forward and side scatter and further staining for nuclear DNA content with Hoechst 33342 dye. |
|---|---|
| Instrument | Flow cytometric analysis or sorting were performed using BD FACSCanto and Sony SH800 Cell Sorter instruments. |
| Software | FlowJo 10.8.0 software. |

| | |
|---|---|
| Cell population abundance | Bladder and prostates from eight- to twelve-week-old male C57BL/6 mice were dissociated into single cells and were stained with antibodies for fluorescence-activated cell sorting on a Sony SH800 Cell Sorter. Bladder urothelial and prostate epithelial cells were sorted and collected based on a Lin(-) CD49f(high) EpCAM(high) immunophenotype. For prostate polymorphic giant cells were analyzed based on forward and side scatter and further staining for nuclear DNA content with Hoechst 33342 dye. Cell population abundance is shown in associated flow plots showing the gating strategy. |
| Gating strategy | Stained prostate and bladder epithelial cells were plotted based on side scatter-area (SSC-A) and FITC (lineage markers) and the lineage-negative population was selected for gating by CD49f-PE and EpCAM-APC. The CD49f(high) EpCAM(high) population was sorted for experimental use.

Dissociated cells from mixed prostate adenocarcinoma an pleomorphic giant cell carcinoma tumors were plotted based on forward scatter-height (FSC-H) and forward scatter-area (FSC-A) to select single cells. Single cells were then gated based on high SSC-A and high FSC-A to isolate larger cells. After passage in organoid culture, dissociated and stained tumor cells were plotted based on SSC-A and Hoechst 33342 staining. Tumor cells were sorted based on high and low Hoechst 33342 staining.

The gating strategies for isolating prostate and bladder epithelial cells and the polymorphic giant cells isolation are provided in the Extended Data Figs. 1a, and 8c, respectively. |

☒ Tick this box to confirm that a figure exemplifying the gating strategy is provided in the Supplementary Information.

