## [Peer Review File · Nature Genetics]

Peer Review Information

Manuscript Title: A combinatorial genetic strategy for exploring complex genotype-phenotype associations in cancer

Corresponding author name(s): Dr John Lee

Reviewer Comments & Decisions:

Decision Letter, initial version:

19th Jul 2023

Dear Dr Lee,

First, please accept my apologies for the delay in returning this decision to you. Thank you so much for bearing with me.

Your Brief Communication, "Combinatorial genetic strategy accelerates the discovery of cancer genotype-phenotype associations" has now been seen by 3 referees. You will see from their comments below that while they find your work of interest, some important points are raised. We are interested in the possibility of publishing your study in Nature Genetics, but would like to consider your response to these concerns in the form of a revised manuscript before we make a final decision on publication.

You'll see that Reviewer #1 has concerns about the overall level of novelty afforded by the work. When revising your manuscript, please could you highlight the advance your work represents over prior studies and include a more in-depth narrative of the potential of the technique in your Discussion. Please bear in mind that Reviewer #1 sits squarely within your target audience so it will be important to convince them of the value of the work. All the comments from Reviewers #2 and #3 should be addressed in full.

We therefore invite you to revise your manuscript taking into account all reviewer and editor comments. Please highlight all changes in the manuscript text file. At this stage we will need you to upload a copy of the manuscript in MS Word .docx or similar editable format.

*2) If you have not done so already please begin to revise your manuscript so that it conforms to our Brief Communication format instructions, available [here](http://www.nature.com/ng/authors/article_types/index.html). Refer also to any guidelines provided in this letter.

[redacted]

We can be flexible about timeframes but please let us know if you anticipate your revisions taking more than 6 months.

Sincerely,

Safia Danovi

Editor
Nature Genetics

Referee expertise:

Referee #1: prostate/bladder cancer models

Referee #2: cancer models, functional genomics

Referee #3: cancer models

Reviewers' Comments:

Reviewer #1:

Remarks to the Author:

The study by Li et al claims to investigate cooperative mechanisms in cancer development using a novel bar coding approach. In principal, this should be an interesting application of new technologies. However, the study fails to do so and largely incomplete.

Major concerns include the following:

1. Overall, the work is descriptive, mechanistic insights are mostly lacking, as is the validation to human cancer (mouse models are investigated).
2. Overall, the study is vague - the abstract in particular does not actually describe a cancer focus or purpose.
3. The Novelty is limited - how does this study/methodology differ from or advance beyond work done by Owen Witte's lab (which used human not mouse tissues)?
4. Related to point #3, the novelty of the bar coding method is unclear as well.
5. A major concern is the overall vagueness of the study and the way it is written.
6. Overall, the rationale for focusing this study on bladder and prostate is unclear.

Reviewer #2:

Remarks to the Author:

The authors describe an innovative approach based on the optimized transduction of mouse primary epithelial cells with barcoded lentiviral libraries of oncogenic events to identify cancer genes that cooperate to transform the target cells into new multifactorial genetic models of cancer.

The manuscript is very well written, and the scope and approach are clear, novel and compelling and would be of interest to the cancer community, especially in the field of novel mouse model development.

I do have some minor comments regarding the genetic characterization of the established models and their comparability to their human counterparts that, if addressed, I believe would further improve the relevance of this study.

1. The authors identify sets of cancer genes whose combined alterations result in specific types of cancer histology (e.g. Fgfr3 S243C,116 Ywhaz, Pik3ca E545K, Pparg, and Pvr14 produce papillary urothelial carcinoma with inverted growth pattern. How frequently do the same gene alterations co-occur in human cancers of the same histology?
2. Since combinatorial genetics is described as the major innovation of the study, I was expecting to see single-cell DNA amplicon sequencing results and deconvolution of the oncogenic events associated with each one of the tumors in the FHBT series. Please explain why the data is not available/not presented, and why the study pivoted to focus on transcriptional profiling instead.
3. The conclusion that the FHIB models are relevant because they 'occupy overlapping space' in the PCA plots of the TCGA BLCA cohort and BBN bladder mouse tumors requires some more detailed explanation. It would be significant to comment on the types/frequencies of cancer gene alterations shared between mouse and human cancers of the same histology and/or transcriptional subtypes.

Minor points

4. Please clarify the nature of the different Mission Bio Tapestry custom single cell amplicon panels: the one described in Extended Data Fig. 1c vs. the one used for the detection of the cancer gene LV Barcodes in Fig. 1f? Is the first one just a test panel to confirm that individual BCs can be detected with high resolution in the premixed 3T3 cell populations? How does it relate to the fig. 1f panel?
5. Please explain how the histological subtype assignments in Fig. 2e are made with reference to the corresponding histological subtype mixture as reported in Fig. 2a? Is it based on the prevalent subtype?

Reviewer #3:

Remarks to the Author:

The manuscript submitted authored by Li, Wong and Sun, et al., presents a novel methodology for transforming and barcoding mouse primary cells, enabling the generation of bladder and prostate cancer models with clinical relevance when injected into immunodeficient mice. The team has conducted an in-depth investigation into the role of commonly mutated genes observed in human bladder and prostate cancers, with a particular focus on elucidating the specific combination of mutations necessary for tumorigenesis.

I find this approach to be exceptionally intelligent and intriguing, and the manuscript effectively outlines the technique employed. However, I believe the team should have no difficulty addressing the following comments, which aim to enhance the clarity and precision of the study.

Major points:

In Extended Data 1a, it would be informative to show all the plots for bladder and prostate tissues. Currently, only the isolation plot for one tissue is shown, while two different tissues were used. To maintain consistency with panel b, where organoids from both tissues are depicted after isolation, it would be convenient to showcase both bladder and prostate isolation plots.

In Extended Data 2c, the relative expression of target genes is displayed by qPCR, but only the downregulated genes using shRNA are presented. It would be valuable to also exhibit the upregulation of those genes that are overexpressed (GOF) to facilitate comparison with the actual expression levels observed in human tumours.

Figure 1d lacks the inclusion or mention of Ctrl grafts in mice using solely embryonic bladder and urogenital mesenchymal cells. It is important to clarify whether this control experiment was performed.

In Figure 1, it would be beneficial to include information regarding the number of mice and tumours obtained, as well as the efficiency of tumour formation.

In Extended Figure 6b, the authors state that their FHBT tumour model exhibits RNA expression patterns resembling those of human cancers from TCGA-BLCA and nitrosamine-induced mouse cancer models. However, the PCA plot demonstrates significant differences between them. This disparity should be addressed and explained more comprehensively.

Minor points:

In Extended Figure 7, tumour cells isolated from tumours were propagated in vitro using organoids to increase cell numbers and analyse mutation burden. However, the authors do not mention the number of passages given, which could potentially impact the mutational landscape. It is crucial to consider that culture conditions might affect mutations within the organoids.

In the text (line 200), there appears to be a typographical error where "urogenital sinus mesenchyme" is duplicated.

In the text (line 227), it would be valuable to provide information on the duration it takes for the tumours to reach 1cm. Additionally, it should be clarified whether this timeframe is dependent on the tumour type and tissue.

Author Rebuttal to Initial comments

RESPONSE TO REVIEWERS' COMMENTS

Reviewer #1 (Remarks to the Author):

The study by Li et al claims to investigate cooperative mechanisms in cancer development using a novel barcoding approach. In principal, this should be an interesting application of new technologies. However, the study fails to do so and largely incomplete.

Major concerns include the following:

Comment 1: Overall, the work is descriptive, mechanistic insights are mostly lacking, as is the validation to human cancer (mouse models are investigated).

Thank you very much for your time and effort in reviewing our manuscript. We have submitted this manuscript as a Brief Communication given our belief that these initial results validate the utility of this novel combinatorial genetics approach to rapidly 1) generate diverse cancer models including clinically relevant bladder cancer and prostate cancer histologies that have not been recapitulated in available genetically engineered mouse models and 2) interrogate genotype-phenotype relationships. We agree that deep mechanistic insights are lacking in this initial report, but additional studies are ongoing and already yielding exciting results that we intend to publish in the future.

Comment 2: Overall, the study is vague - the abstract in particular does not actually describe a cancer focus or purpose.

Thank you for this comment. The abstract of a Brief Communication is limited to three sentences/70 words and we tried our best to succinctly convey the most salient points of the manuscript within this constraint. To further clarify the purpose of the study, we have now significantly amended the first paragraph of the main text in lines 48-69 to elaborate on the cancer focus.

Comment 3: The Novelty is limited - how does this study/methodology differ from or advance beyond work done by Owen Witte's lab (which used human not mouse tissues)?

Thank you for allowing clarification of this point. We have added text to lines 50-56 and 57-64 to describe potential shortcomings of available cancer models and current genetically-defined methodologies employed in functional cancer genomics. We also mention the need for advances in “scale, throughput, and economy” on line 65 relative to established genetically-engineered mouse models and dissociated-cell tissue recombination/transplantation assays for cancer. In this context, we describe how the methodology described may overcome prior limitations and provide a clearer basis for novelty.

The studies performed by Dr. Witte's laboratory were reliant on a low-throughput approach with parallel testing of combinations of defined sets of genetic insults (i.e., each graft receives either genes A+B or A+C or B+C or A+B+C+D). A key advancement of our study is the ability to introduce—at high efficiency and throughput (as shown in Fig. 1b and 1c)—many combinations of gain- or loss-of-function genetic events simultaneously from a barcoded lentiviral library within each graft (i.e., genes A-Z where individual cells within the graft receive diverse gene combinations A+B+C, B+E+G+H+Z, A+C+R+X+Y+Z, etc.). Further, we have implemented custom, highly sensitive, targeted single-cell DNA amplicon sequencing in a novel manner to enable massively parallel deconvolution of multiple barcoded lentiviruses integrated into individual cells within a tumor. This allows us to track the clonal architecture of the tumor and associate genetic interactions that may have promoted tumorigenesis and specific tumor

phenotypes.

We have added language to the manuscript in lines 183-188 to further highlight the value of this methodology in our proof-of-concept studies: “We leveraged this strategy to develop a series of mouse bladder cancers that recapitulate the phenotypic diversity of human bladder cancer and a mouse prostate cancer with pleomorphic

giant cell carcinoma, representing cancer subtypes that have not previously been modeled in a genetically- defined fashion. Importantly, single-cell LV barcode deconvolution associated mutant active Pparg with luminal papillary differentiation of urothelial carcinoma and loss of Kmt2c with pleomorphic giant cell carcinoma in prostate cancer.”

Comment 4: Related to point #3, the novelty of the bar-coding method is unclear as well.

We have addressed the novelty of the barcoding method and single-cell DNA amplicon sequencing strategy in our response to Comment 3.

Comment 5: A major concern is the overall vagueness of the study and the way it is written.

We have made changes to the manuscript text as described in responses to Comments #2 and #3 to better convey how this study represents a scientific advance and the relative importance of our results.

Comment 6: Overall, the rationale for focusing this study on bladder and prostate is unclear.

We focused on bladder and prostate cancer given our prior experience isolating primary mouse epithelial cells from these tissues and our cancer-specific expertise in this area. Specifically related to bladder cancer, available genetically-engineered mouse models have poorly reflected the diversity of the human disease and so our intention was to seize this as an opportunity to show that we could recapitulate diverse bladder cancer phenotypes including variant histologies by employing this combinatorial genetic approach. While the manuscript focuses on bladder and prostate cancer for proof-of-concept studies, we believe that this approach may be applicable to and have broad utility for the study of many cancer types and other polygenic diseases.

Reviewer #2 (Remarks to the Author):

The authors describe an innovative approach based on the optimized transduction of mouse primary epithelial cells with barcoded lentiviral libraries of oncogenic events to identify cancer genes that cooperate to transform the target cells into new multifactorial genetic models of cancer.

The manuscript is very well written, and the scope and approach are clear, novel and compelling and would be of interest to the cancer community, especially in the field of novel mouse model development.

I do have some minor comments regarding the genetic characterization of the established models and their comparability to their human counterparts that, if addressed, I believe would further improve the relevance of this study.

Comment 1: The authors identify sets of cancer genes whose combined alterations result in specific types of cancer histology (e.g. Fgfr3 S243C, Ywhaz, Pik3ca E545K, Pparg, and Pvr14 produce papillary urothelial carcinoma with inverted growth pattern. How frequently do the same gene alterations co-occur in human cancers of the same histology?

Thank you very much for your careful consideration of our manuscript. We agree that it is important to include information regarding the co-occurrence of these alterations in human bladder cancer. We have now added Extended Data Fig. 5 (shown below) which is an OncoPrint from cBioPortal showing the frequencies and types of gene alterations associated with human muscle-invasive bladder cancer from the TCGA-BLCA cohort. We focus specifically on the gain- and loss-of-function alterations encoded in our mBU-LVp including Pparg, Pvr14, Ywhaz, Pik3ca, and Fgfr3. While the types of histologies are not annotated in this dataset, it is evident that these gene alterations co-occur. As we have described in the manuscript, FGFR3 activating mutations have previously been shown to be highly enriched in papillary urothelial carcinomas.

Genetic Alteration

Comment 2: Since combinatorial genetics is described as the major innovation of the study, I was expecting to see single-cell DNA amplicon sequencing results and deconvolution of the oncogenic events associated with each one of the tumors in the FHBT series. Please explain why the data is not available/not presented, and why the study pivoted to focus on transcriptional profiling instead.

We agree that it is important to deconvolute the oncogenic events associated with each of the tumors in the FHBT series. In the manuscript, we show deconvolution of one of the FHBT tumors and functional validation of the role of Pparg in driving luminal papillary urothelial carcinoma. We also associate loss of Kmt2c with pleomorphic giant cell carcinoma in a prostate tumor. However, we are actively deconvoluting all of the bladder and prostate tumors and performing functional studies to understand the key genetic interactions that may be critical in driving each cancer phenotype. As you can imagine, this work will take some time and is already

yielding exciting results that we intend to publish in the future. Our intention was to submit this initial manuscript as a proof-of-concept of the technology and its potential utility. We focused on transcriptional profiling for more robust phenotypic validation of the clinical relevance of the FHBT models.

Comment 3: The conclusion that the FHIB models are relevant because they ‘occupy overlapping space’ in the PCA plots of the TCGA BLCA cohort and BBN bladder mouse tumors requires some more detailed explanation. It would be significant to comment on the types/frequencies of cancer gene alterations shared between mouse and human cancers of the same histology and/or transcriptional subtypes.

We have updated the PCA projection plots in Fig. 2i and Extended Data Fig. 7b (shown below) after incorporating batch correction using ComBat-seq between the datasets. We have now included 90% confidence ellipses for each histology and/or consensus classification to enhance the visual interpretation of the results. We have also added clarification in the text on lines 161-163: “...occupy overlapping space based on histologic classification, indicating that the transcriptional features with the greatest variance between tumor subtypes are also conserved with FHBT models.”

As mentioned, we have added an Oncoprint to Extended Data Fig. 5 to show specifically genes included in our mBU-LVp and their alteration frequencies in human muscle-invasive bladder cancer from the TCGA-BLCA cohort. Unfortunately, the overwhelming majority of TCGA-BLCA cases are annotated as “urothelial carcinoma” and there are only a handful of variant histologic subtypes within the cohort. Pathologists often do not note variant histologic subtypes of bladder cancer unless it is the dominant cancer histology. Thus, the small number of such annotated cases with cancer genome sequencing makes it currently difficult to answer the frequencies of gene alterations shared between mouse FHBT models and human bladder cancers of the same histologic subtypes.

Minor points

Comment 4: Please clarify the nature of the different Mission Bio Tapestri custom single cell amplicon panels: the one described in Extended Data Fig. 1c vs. the one used for the detection of the cancer gene LV Barcodes in Fig. 1f? Is the first one just a test panel to confirm that individual BCs can be detected with high resolution in the premixed 3T3 cell populations? How does it relate to the fig. 1f panel?

We apologize that this was not clear. Extended Data Fig. 1c describes the Mission Bio Tapestri custom single-cell amplicon sequencing panel that was applied in the experiments shown in Fig. 1f and Extended Data Fig. 1d. The experimental data in Extended Data Fig. 1d is shown to confirm the ability of the panel to accurately

and sensitively deconvolute the clonal architecture of a well-defined set of 3T3 cells with

mixtures of clones harboring different combinations of lentiviral barcodes. After validating this technology, we then applied the custom single-cell amplicon sequencing panel to our dissociated tumor as shown in Fig. 1f.

Comment 5: Please explain how the histological subtype assignments in Fig. 2e are made with reference to the corresponding histological subtype mixture as reported in Fig. 2a? Is it based on the prevalent subtype?

Yes, the histological subtype assignment in Fig. 2e was made based on the prevalent subtype.

Reviewer #3 (Remarks to the Author):

The manuscript submitted authored by Li, Wong and Sun, et al., presents a novel methodology for transforming and barcoding mouse primary cells, enabling the generation of bladder and prostate cancer models with clinical relevance when injected into immunodeficient mice. The team has conducted an in-depth investigation into the role of commonly mutated genes observed in human bladder and prostate cancers, with a particular focus on elucidating the specific combination of mutations necessary for tumorigenesis.

I find this approach to be exceptionally intelligent and intriguing, and the manuscript effectively outlines the technique employed. However, I believe the team should have no difficulty addressing the following comments, which aim to enhance the clarity and precision of the study.

Major points:

Comment 1: In Extended Data 1a, it would be informative to show all the plots for bladder and prostate tissues. Currently, only the isolation plot for one tissue is shown, while two different tissues were used. To maintain consistency with panel b, where organoids from both tissues are depicted after isolation, it would be convenient to showcase both bladder and prostate isolation plots.

Thank you very much for these excellent comments. As suggested, we have now included representative bladder (top) and prostate (bottom) isolation plots in Extended Data Fig. 1a (shown below).

Comment 2: In Extended Data 2c, the relative expression of target genes is displayed by qPCR, but only the downregulated genes using shRNA are presented. It would be valuable to also exhibit the upregulation of those genes that are overexpressed (GOF) to facilitate comparison with the actual expression levels observed in human tumours.

Thank you for this suggestion. We have validated the upregulation of the overexpressed (gain-of-function) genes in 3T3 cells by qPCR. This data is now plotted in Extended Data Fig. 2d (shown below). We have also added the identities of the primers used for these studies to Extended Data Table 3.

Comment 3: Figure 1d lacks the inclusion or mention of Ctrl grafts in mice using solely

embryonic bladder and urogenital mesenchymal cells. It is important to clarify whether this control experiment was performed.

We appreciate this comment. We did add this key control in our experiments and none of these grafts formed tumors. We have now clarified this in line 114-115 of the text: “No tumors were appreciable from control grafts of untransduced mBU or mPE cells recombined with EBLM or UGSM.”

Comment 4: In Figure 1, it would be beneficial to include information regarding the number of mice and tumours obtained, as well as the efficiency of tumour formation.

We agree that this is beneficial and important information. We have now included Extended Data Table 2 (shown below) that summarizes the efficiency of tumor formation. We have added a description of this data to the text on lines 115-117: “The efficiency of tumor formation (tumors formed per graft inoculated) was 80% (16 of 20) for mBU cells infected with BU-LVp and 38% (18 of 47) for mPE cells infected with PE-LVp (Extended Data Table 2).”

Tissues	Number of mice inoculated	Tumor incidence (%)	Average time to 1 cm tumor diameter (months)
Bladder	20	80%	4.2
Prostate	47	38%	8.9

Comment 5: In Extended Figure 6b, the authors state that their FHBT tumour model exhibits RNA expression patterns resembling those of human cancers from TCGA-BLCA and nitrosamine-induced mouse cancer models. However, the PCA plot demonstrates significant differences between them. This disparity should be addressed and explained more comprehensively.

We appreciate this comment. We did not previously apply batch correction but after using ComBat-seq, we find that the PCA projections now show greater similarity between tumor subtypes comparing FHBT to TCGA- BLCA and BBN tumors. Our response to Comment 3 from Reviewer 2 shows these new plots and our additional description of these results.

Minor points:

Comment 6: In Extended Figure 7, tumour cells isolated from tumours were propagated in vitro using organoids to increase cell numbers and analyse mutation burden. However, the authors do not mention the number of passages given, which could potentially impact the mutational landscape. It is crucial to consider that culture conditions might affect mutations within the organoids.

We propagated the cells for one passage to increase cell numbers for further downstream analysis. This information has been provided in the revised manuscript on line 172.

Comment 7: In the text (line 200), there appears to be a typographical error where "urogenital sinus mesenchyme" is duplicated.

Thank you for catching this typographical error. This has been corrected on line 233 to replace one of the "urogenital sinus mesenchyme" with "embryonic bladder mesenchyme."

Comment 8: In the text (line 227), it would be valuable to provide information on the duration it takes for the tumours to reach 1cm. Additionally, it should be clarified whether this timeframe is dependent on the tumour type and tissue.

We have now included Extended Data Table 2 (shown above in response to Comment 4) that summarizes tumor latency. We have added a description of this data to the text on lines 117-120: "Tumor latency was measured as time from inoculation to achieving a maximal tumor diameter of 1 cm and ranged from 2.3-7.4 months (mean 4.2 months) for bladder tumors and 3.2-16 months (mean 8.9 months) for prostate tumors (Extended Data Table 2)." Indeed, we did appreciate a difference in tumor latency as well as tumor initiation frequency based on tissue type. However, it is unclear whether this difference is related to tissue type alone because this is complicated by the bladder or prostate cancer-specific oncogenic insults encoded in each lentiviral library.

Decision Letter, first revision:

10th Nov 2023

Dear Dr. Lee,

Thank you for submitting your revised manuscript "Combinatorial genetic strategy accelerates the discovery of cancer genotype-phenotype associations" (NG-BC62414R). It has now been seen by the original referees and their comments are below. The reviewers find that the paper has improved in revision, and therefore we'll be happy in principle to publish it in Nature Genetics, pending minor revisions to comply with our editorial and formatting guidelines.

Sincerely,

Safia Danovi
Editor
Nature Genetics

Reviewer #1 (Remarks to the Author):

This Reviewer's concerns about the appropriateness of the study for Nature Genetics have not been addressed in this revision. This Reviewer's impression is that the study is lacking conceptual and technical innovation that would be appropriate for Nature Genetics. There is also no take-home message or any real mechanistic insights.

A "brief" communication should not mean an incomplete study.

Reviewer #2 (Remarks to the Author):

The authors adequately replied to my requests. I have no additional comments.

Reviewer #3 (Remarks to the Author):

The authors have adequately addressed all the concerns I raised regarding this study, and I am pleased to recommend this manuscript for publication in Nature Genetics.

Author Rebuttal, first revision:

Point-by-point response

Reviewer #1:

Remarks to the Author:

This Reviewer's concerns about the appropriateness of the study for Nature Genetics have not been addressed in this revision. This Reviewer's impression is that the study is lacking conceptual and technical innovation that would be appropriate for Nature Genetics. There is also no take-home message or any real mechanistic insights.

A "brief" communication should not mean an incomplete study.

Response: We respect the reviewer's comments which led us to make constructive changes to the manuscript text with the previous revision to better convey the conceptual and technical innovation of the work to a general audience. We believe that this has improved the quality of the manuscript.

Reviewer #2:

Remarks to the Author:

The authors adequately replied to my requests. I have no additional comments.

Response: Thank you.

Reviewer #3:

Remarks to the Author:

The authors have adequately addressed all the concerns I raised regarding this study, and I am pleased to recommend this manuscript for publication in Nature Genetics.

Response: Thank you.

Final Decision Letter:

25th Jan 2024

Dear Dr Lee,

I am delighted to say that your manuscript "A combinatorial genetic strategy for exploring complex genotype-phenotype associations in cancer" has been accepted for publication in an upcoming issue of Nature Genetics.

Due to the importance of these deadlines, we ask that you please let us know now whether you will be difficult to contact over the next month. If this is the case, we ask you provide us with the contact

information (email, phone and fax) of someone who will be able to check the proofs on your behalf, and who will be available to address any last-minute problems.

Your paper will be published online after we receive your corrections and will appear in print in the next available issue. You can find out your date of online publication by contacting the Nature Press Office (press@nature.com) after sending your e-proof corrections.

Please note that *Nature Genetics* is a Transformative Journal (TJ). Authors may publish their research with us through the traditional subscription access route or make their paper immediately open access through payment of an article-processing charge (APC). Authors will not be required to make a final decision about access to their article until it has been accepted. [Find out more about Transformative Journals](https://www.springernature.com/gp/open-research/transformative-journals)

Authors may need to take specific actions to achieve [compliance with funder and institutional open access mandates](https://www.springernature.com/gp/open-research/funding/policy-compliance-faqs). If your research is supported by a funder that requires immediate open access (e.g. according to [Plan S principles](https://www.springernature.com/gp/open-research/plan-s-compliance)) then you should select the gold OA route, and we will direct you to the compliant route where possible. For authors selecting the subscription publication route, the journal's standard licensing terms will need to be accepted, including [those licensing terms will supersede any other terms that the author or any third party may assert apply to any version of the manuscript](https://www.nature.com/nature-portfolio/editorial-policies/self-archiving-and-license-to-publish).

If you have posted a preprint on any preprint server, please ensure that the preprint details are

updated with a publication reference, including the DOI and a URL to the published version of the article on the journal website.

If you have not already done so, we invite you to upload the step-by-step protocols used in this manuscript to the Protocols Exchange, part of our on-line web resource, natureprotocols.com. If you complete the upload by the time you receive your manuscript proofs, we can insert links in your article that lead directly to the protocol details. Your protocol will be made freely available upon publication of your paper. By participating in natureprotocols.com, you are enabling researchers to more readily reproduce or adapt the methodology you use. [Natureprotocols.com](http://natureprotocols.com) is fully searchable, providing your protocols and paper with increased utility and visibility. Please submit your protocol to <https://protocolexchange.researchsquare.com/>. After entering your nature.com username and password you will need to enter your manuscript number (NG-BC62414R1). Further information can be found at <https://www.nature.com/nature-portfolio/editorial-policies/reporting-standards#protocols>

Sincerely,

Safia Danovi
Editor
Nature Genetics